# Exploring the Spatial Autocorrelation in Soil Moisture Networks: Analysis of the Bias from Upscaling the Texas Soil Observation Network (TxSON)

**Yaping Xu** [1] , **Cuiling Liu** [2,*], **Lei Wang** [3] **and Lei Zou** [4]

1 Department of Plant Sciences, University of Tennessee, Knoxville, TN 37996, USA
2 Urban Informatics and Shenzhen Key Laboratory of Spatial Smart Sensing and Services, Shenzhen University, Shenzhen 518060, China
3 Department of Geography & Anthropology, Louisiana State University, Baton Rouge, LA 70803, USA
4 Department of Geography, Texas A&M University, College Station, TX 77843, USA
* Correspondence: cuiling_liu@szu.edu.cn

**Abstract:** Microwave remote sensing such as soil moisture active passive (SMAP) can provide soil moisture data for agricultural and hydrological studies. However, the scales between station-measured and satellite-measured products are quite different, as stations measure on a point scale while satellites have a much larger footprint (e.g., 9 km). Consequently, the validation for soil moisture products, especially inter-comparison between these two types of observations, is quite a challenge. Spatial autocorrelation among the stations could be a contribution of bias, which impacts the dense soil moisture networks when compared with satellite soil moisture products. To examine the effects of spatial autocorrelation to soil moisture upscaling models, this study proposes a spatial analysis approach for soil moisture ground observation upscaling and Thiessen polygon-based block kriging (TBP kriging) and compares the results with three other methods typically used in the current literature: arithmetic average, Thiessen polygon, and Gaussian-weighted average. Using the Texas Soil Observation Network (TxSON) as ground observation, this methodology detects spatial autocorrelation in the distribution of the stations that exist in dense soil moisture networks and improved the spatial modeling accuracy when carrying out upscaling tasks. The study concluded that through TBP kriging the minimum root-mean-square deviation (RMSD) is given where spatial autocorrelation takes place in the soil moisture stations. Through TBP kriging, the station-measured and satellite-measured soil moisture products are more comparable.

**Keywords:** soil moisture upscaling; soil moisture active and passive (SMAP); TxSON; soil moisture network; spatial autocorrelation; block kriging; Thiessen polygon





## 1. Introduction

Soil moisture plays an important role in both atmospheric circulation and the water cycle. Soil moisture is the key component of understanding global and regional climate change and its impact on agriculture and hydrological applications. Currently, drought prediction relies more heavily on observing the moisture levels in the atmosphere (i.e., humidity) than the moisture levels of soil [1]; yet this is mostly due to the lack of soil moisture data availability. Having continuous soil moisture measurements will improve crop yield forecasting and irrigation planning. The status of the soil is also vital to hydrological studies. When the soil is saturated, seepage and infiltration are barely possible, triggering surface runoff, which will increase the potential for flooding to occur. Knowing the soil status, especially the soil moisture status, is critical to reducing such disasters.

Typically, three methods can be used to measure soil moisture: station, satellite, and land surface model simulation. Generally, in situ soil moisture measurement methods (e.g., gravimetric, TDR, etc.) are observed at a scale of meters (point scale), while satellite

measurements are observed at a scale of several square kilometers (pixel size) [2]. In other words, soil moisture observations from the satellite and ground are inconsistent in their scales. Studies showed that soil moisture varies significantly from small scales (<10 m) to field scales and larger (>1 km) scales [3]. Current statistical models that tried to link these two types of observations directly in model calibration usually generate calibrated models under the condition of a modifiable area unit problem (MAUP): the statistical models calibrated at one scale do not usually work at other scales when individual samples (ground) are aggregated (satellite).

There are two common approaches to matching the scales between satellite and station soil moisture datasets: downscaling [4–7] and upscaling [8–10]. Soil moisture downscaling aims to downscale the soil moisture products from satellites. This operation can bring down the soil moisture from a coarse spatial resolution (i.e., ~10 km) to a medium spatial resolution (~1 km, typically), which significantly decreases the bias caused by the scale difference between the two types of observations [7]. This approach usually involves SMAP soil moisture product evaluation with ground-based in situ soil moisture observations. However, as a station can only measure the soil moisture at a local scale of square meters, it does not represent the large footprint measured from the satellite, even if it is downscaled to a 1 km scale. Direct comparison between these two types of measurement will oftentimes cause large bias.

Upscaling involves the operation that scales up the points level to match the satellite grid. This approach is very different from the downscaling, so, rather than use the point-based in situ soil moisture stations to calibrate SMAP directly, soil moisture upscaling uses models or algorithms to scale up the point to an aerial observation, and is compared with SMAP soil moisture observation as the target to get a mutual validation between the satellite observation and upscaled models. The challenge is in examining whether a soil moisture network meets the measurement requirement. Soil moisture networks can be categorized into dense networks and sparse networks. A sparse network means typically one to two ground observations per satellite footprint [11], whereas a dense or core network consists of dozens of ground observations per satellite footprint [9,12].

Spatial autocorrelation is a term that describes the systematic spatial variation in a single variable on a two-dimensional surface [13]. Spatial autocorrelation refers to the correlation among the values of that variable strictly attributable to their distances. Where adjacent observations have similar data values, positive spatial autocorrelation is said to occur; where adjacent observations tend to have very contrasting values, then negative spatial autocorrelation is said to occur [14]. Spatial autocorrelation deals with two distinct types of information: attributes of the spatial feature—soil moisture, as well as the spatial feature of the location itself—the position on a map [15]. Choosing an optional upscaling strategy is not an easy task, especially when spatial autocorrelation exists in the dataset [15–19]. In the case of dense networks, for example, the Texas Soil Observation Network (TxSON), spatial autocorrelation exists in the dataset, and therefore special techniques might be necessary to upscale the dataset.

Commonly used upscaling strategies in the literature include four types. The first strategy is using a time-stability concept, which means locating the soil moisture stations at 'representative' landscape locations [9] and predicting large-scale moisture averages from these few sensors located at particular sites. It was originally developed to sample a 2000 m$^2$ sparsely instrumented grass field based on stable measurement sites that predict the large-scale average over long time scales [20] and later extended to a larger network [21]. This strategy requires static vegetation type, soil type, and topography [21–24]. It is reported that only six points are required within a 75 km$^2$ footprint of a satellite [25]. However, a great challenge is that direct identification of time-stable sites typically requires very dense spatial sampling of a coarse-scale area over an extended period. The second strategy is with spatial analytical models that include arithmetic averages [26], the Thiessen polygon, also as known as the Voronoi upscaling method [26–28], inverse distance-weighted interpolation [29,30], and kriging [31,32]. The fundamental difference among these methods lies in the approach

of how to assign proper weight to the stations to obtain an accurate estimate of the average soil moisture. With regard to spatial up-scaling, unless the in situ measurements are dense and evenly spread across the site, using the arithmetic average of the measured values does not guarantee an accurate estimate of the grid average soil moisture [26]. The third strategy is using land surface models [8,33] and geostatistical models [10,34] such as regression, random forests, etc., which are commonly used for sparse networks. The fourth strategy is based on a field campaign. Intensive soil moisture measurements are collected at different times and stations and therefore create the dense soil moisture network. However, this strategy is very expensive, and is also open to combination with any of the above-mentioned strategies for a better upscaling result. Overall, spatial statistical models are widely used for dense networks because of their accurate estimates of the average soil moisture, and among the available spatial statistical models, the Thiessen polygon was chosen as the default upscaling algorithm for TxSON as part of the calibration/validation procedures [26,35,36].

The hypothesis of this study is that spatial autocorrelation exists in a dense soil moisture network, and when spatial autocorrelation is detected, we need to revisit the TxSON's default algorithm, the Thiessen polygon, to verify if it is an optimized upscaling approach. To address the potential spatial autocorrelation existing in the Thiessen polygon for an optimized upscaling method, this study proposes to use a spatial statistical model, Thiessen polygon-based block kriging (denoted as TBP kriging hereafter), and compare the performance with the original Thiessen polygon method to evaluate its validity. This paper started by examining the spatial autocorrelation in an existing soil moisture network, the Texas Soil Observation Network (TxSON), and confirming the presence of such autocorrelation. Then, a new method that integrates the geometry of the station distribution, as well as the correlation between the stations, is used to handle the spatial autocorrelation. This new method is also compared with three other methods to evaluate its validity when dealing with spatial autocorrelation. Through this TBP kriging method, we expect to reduce the bias inherent in the direct comparison between the two types of soil moisture observations.

It was generally assumed in the literature that the soil moisture data measured in the morning are more stable than in the afternoon from the radar or radiometer because the land surface temperature and soil dielectric properties are likely to be more uniform in the vertical profiles of soil [37–41]; however, further studies are required to confirm this hypothesis through experiments. In this study, the morning/afternoon data obtained from soil moisture active passive (SMAP) will be compared to discover the effects of spatial autocorrelation on soil moisture upscaling performance.

## 2. Material and Methods

### 2.1. Datasets and Study Area

The main datasets in this study include satellite data acquired from NASA's SMAP (Jet Propulsion Laboratory, La Cañada Flintridge, CA, USA; NASA Goddard Space Flight Center, Greenbelt, MD, USA) satellite, and ground station data collected from the TxSON (Fredericksburg, TX, USA).

The SMAP data products were collected as Level 3 soil moisture data (SMAP L3-SM-P-E, NASA, 2017) from an L-band radiometer (Jet Propulsion Laboratory, La Cañada Flintridge, CA, USA; NASA Goddard Space Flight Center, Greenbelt, MD, USA). The Level 3 data were a daily global composite of Level 2 data, a soil moisture product based on brightness temperature measurements that were sensitive to soil moisture. SMAP Level 3 data measure the top 5 cm of the soil column daily at 6 AM/6 PM local solar time [38]. The spatial resolution was 9 km per satellite pixel, and the measuring accuracy was equal to or better than 0.04 cm$^3$/cm$^3$ [38]. SMAP Level 3 products were downloaded from NASA's Earth Observing System Data and Information System (EOSDIS) Reverb Echo portal on EARTHDATA as Geo Tagged Image File Format (GeoTIFF) in the World Geodetic System (WGS) 1984 Geographic Coordinate System.

TxSON is an intensively monitored 36 km (1300 km$^2$) grid-cell soil moisture network specifically designed for calibration and validation of remotely sensed soil moisture esti-

mates. The network consists of 40 monitoring stations that measure in situ soil moisture, soil temperature, and precipitation in real time. The 40 stations include 27 micro-stations, 6 weather stations, and 7 partner stations with the Lower Colorado River Authority (LCRA). The network is located near Fredericksburg, Texas (30.27° N, 98.87° W), along the Pedernales River and within the middle reaches of the lower Colorado River. TxSON uses a nested design to replicate soil moisture at 9 and 36 km satellite pixels to support the SMAP satellite and its Calibration and Validation Program [42]. The 3 km grids were not publicly available when this study was performed. TxSON measures soil moisture at 5 cm, 10 cm, 20 cm, and 50 cm depth with soil moisture sensors horizontally inserted into the soil. The sampling intervals were 5 minutes, which were then averaged and updated hourly. The soil moisture sensors were Campbell CS-655s (Campbell Scientific, Inc., Logan, UT, USA) and their measuring accuracy was reported as 0.043 m³/m³ based on standard factory calibration [35]. Sitewide calibration was done by Calwell et al. [42], and the accuracy was reported as 0.027 m³/m³, better than the factory accuracy of 0.043 m³/m³.

This study used the 9 km grid TxSON data at a depth of 5 cm for the upscaling (Figure 1). The justification for using the 9 km grids, rather than the 36 km grids, was that they match SMAP for a direct comparison: the 9 km grid was the designated resolution and 5 cm was the designated depth of the SMAP Level 3 soil moisture product, while the 36 km grid was larger in the footprint for verification. To match the time interval of SMAP, the hourly dataset from July 1 to September 16, 2018, was collected and then filtered with the temporal resolution of the data daily at 6 AM and 6 PM local time. As TxSON uses local time, Central Daylight Time (CDT) for the summer (UTC-05), and SMAP uses local solar time (UTC-6.6), there is a 1.6-hour time difference between SMAP and TxSON. The specific date range was used because of two reasons: (1) that they were the first long time-series data available when directly requested from the TxSON project, and (2) this time period had less rainfall, which was important for the reliable readings from both the ground station and satellite.

The SMAP L3-SM-P-E 9 km soil moisture datasets were downloaded as an .hdf file, and the extract subdataset tool was used to get the soil moisture layers. They are layer 12 (AM soil moisture) and layer 41 (PM soil moisture). The TxSON data were originally provided in the MATLAB (R2019a) file. We used .csv files to store the soil moisture information for 78 days and extracted the 6 AM/6 PM data.

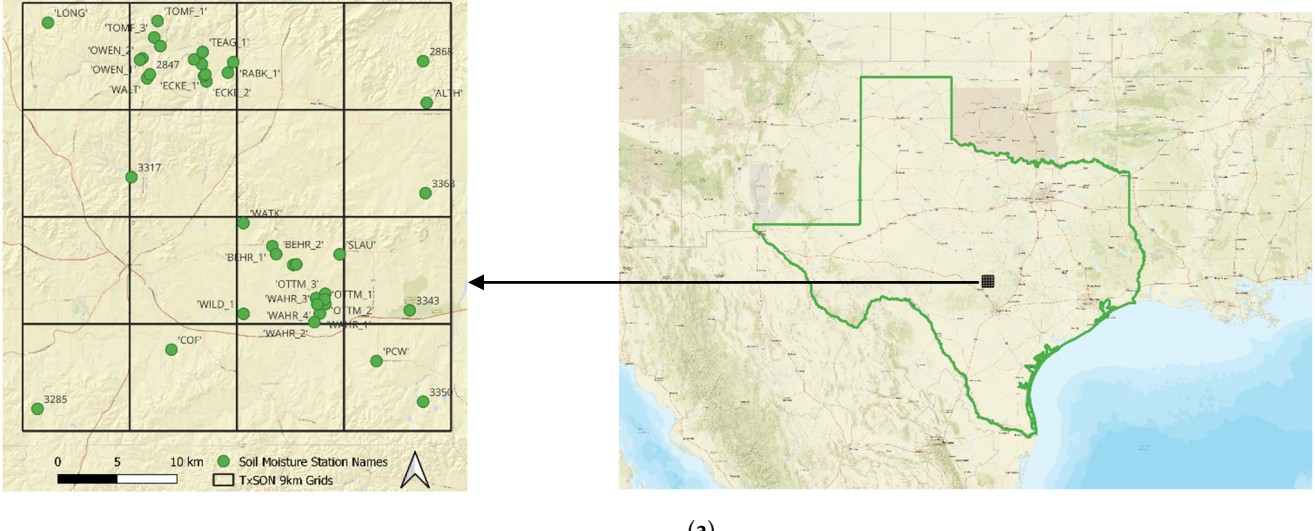

(**a**)

**Figure 1.** *Cont.*

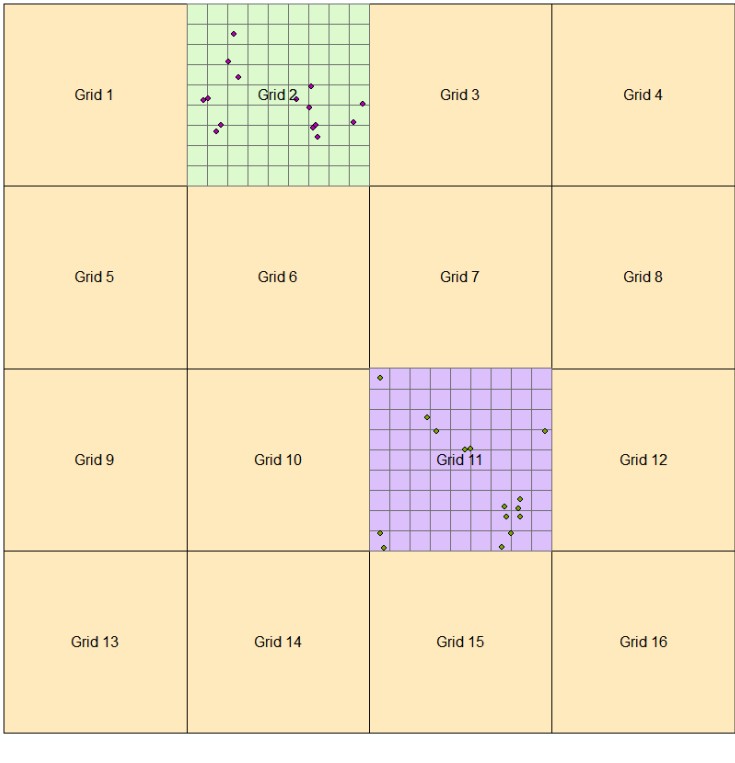

**(b)**

**Figure 1.** Study area and the layout of TxSON. (**a**) Location of the study area and soil moisture stations; (**b**) the station and the grid layout of TxSON. The TxSON soil moisture has 16 grids, ranging from grid 1 to 16, each 9 km by 9 km in size. Grids 2 and 11 are divided into 1 km subgrids. The dimension of the TxSON is 36 km by 36 km. The figure only shows the stations within grids 2 and 11; stations included in other grids were omitted for this study.

Table 1 shows the soil type variation in the study area. The soil units were from the Soil Survey Geographic Database (SSURGO). The vegetation types were obtained from the 2019 version of the National Land Cover Database (NLCD). With this information, the time-stability concept as mentioned in the introduction part was not suitable for the area presented in this study: the validation for the time stability required an extended period that was not available for this study. Land surface models and geostatistical models were also not recommended because they required sparse networks. A field campaign, which involved intensive soil moisture measurements at different times and stations, is more expensive and labor-intensive to carry out. Comparatively, spatial statistical models were suitable for upscaling the TxSON soil moisture data to match the SMAP data, and the methodology presented in this paper falls within this category.

**Table 1.** TxSON Stations within the Study Area.

| Site Name | Soil Unit | F_loggerID | Latitude | Longitude | Land Use Type |
|---|---|---|---|---|---|
| | | Grid 2 | | | |
| TEAG_1 | Bastrop loamy fine sand, 1 to 5 percent slopes | 'CR200_1' | 30.4376 | −98.8059 | Pasture/Hay |
| TEAG_2 | Heaton loamy fine sand | 'CR200_3' | 30.4283 | −98.8065 | Shrub/Scrub |
| RABK_1 | Luckenbach clay loam | 'CR200_4' | 30.4298 | −98.7792 | Shrub/Scrub |
| TEAG_3 | Loneoak fine sand | 'CR200_9' | 30.4319 | −98.8133 | Shrub/Scrub |
| OWEN_1 | Luckenbach clay loam | 'CR200_13' | 30.4327 | −98.8583 | Shrub/Scrub |
| ECKE_2 | Heaton loamy fine sand | 'CR200_14' | 30.4151 | −98.8025 | Shrub/Scrub |

**Table 1.** *Cont.*

| Site Name | Soil Unit | F_loggerID | Latitude | Longitude | Land Use Type |
|---|---|---|---|---|---|
| WALT | Purves soils | 'CR200_19' | 30.4175 | −98.8542 | Shrub/Scrub |
| RABK_2 | Brackett soils | 'CR200_21' | 30.4218 | −98.7839 | Evergreen Forest |
| OWEN_2 | Heaton loamy fine sand | 'CR200_22' | 30.4315 | −98.8604 | Open Space |
| ECKE_3 | Brackett soils | 'CR200_26' | 30.4193 | −98.8046 | Deciduous Forest |
| TOMF_1 | Tarrant soils | 'CR200_28' | 30.4613 | −98.8451 | Shrub/Scrub |
| TOMF_3 | Purves soils | 'CR200_29' | 30.4487 | −98.8480 | Shrub/Scrub |
| ECKE_1 | Krum silty clay | 'CR1000_1' | 30.4205 | −98.8033 | Shrub/Scrub |
| TOMF_2 | Oakalla silty clay loam | 'CR1000_6' | 30.4421 | −98.8427 | Shrub/Scrub |
| 2847 | Purves | 'LCRA_2' | 30.4206 | −98.8519 | Open Space |
| | | Grid 11 | | | |
| BEHR_1 | Luckenbach clay loam | 'CR200_2' | 30.2897 | −98.7462 | Shrub/Scrub |
| WILD_1 | Hensley loam | 'CR200_5' | 30.2381 | −98.7701 | Evergreen Forest |
| WAHR_1 | Tobosa Clay | 'CR200_6' | 30.2383 | −98.7037 | Grassland/Herbaceous |
| WAHR_2 | Bastrop fine sandy loam | 'CR200_7' | 30.2318 | −98.7084 | Pasture/Hay |
| SLAU | Brackett soils | 'CR200_8' | 30.2834 | −98.6864 | Shrub/Scrub |
| WATK | Pedernale fine sandy loam | 'CR200_10' | 30.3072 | −98.7703 | Shrub/Scrub |
| WAHR_3 | Hensley loam | 'CR200_15' | 30.2501 | −98.7069 | Shrub/Scrub |
| BEHR_2 | Oakalla silty clay loam | 'CR200_16' | 30.2836 | −98.7417 | Shrub/Scrub |
| RODE_1 | Purves soils | 'CR200_17' | 30.2754 | −98.7268 | Shrub/Scrub |
| OTTM_1 | Brackett soils | 'CR200_18' | 30.2456 | −98.6988 | Shrub/Scrub |
| OTTM_3 | Hensley loam | 'CR200_24' | 30.2534 | −98.6990 | Shrub/Scrub |
| OTTM_2 | Bastrop loamy fine sand | 'CR200_25' | 30.2492 | −98.6995 | Open Space |
| WAHR_4 | Pedernale fine sandy loam | 'CR1000_2' | 30.2454 | −98.7059 | Shrub/Scrub |
| RODE_2 | Krum silty clay | 'CR1000_3' | 30.2758 | −98.7242 | Shrub/Scrub |

*2.2. Methodology Overview*

As our general hypothesis is that when spatial autocorrelation exists in the soil moisture network, the commonly used upscaling algorithm, such as the Thiessen polygon, will need to be optimized to consider the spatial autocorrelation. Therefore, to begin with, we used a spatial autocorrelation detection method, Moran's I, to detect whether or not spatial autocorrelation exists in the network. We provided two scenarios here based on the network settings of TxSON: gird 2 detected spatial autocorrelation, and grid 11 did not detect spatial autocorrelation. For both scenarios, we used three soil moisture upscaling common models: Thiessen polygon, arithmetic average, and Gaussian-weighted average to upscale the TxSON soil moisture observations over a course of 78 days, 1 July to 16 September 2018. As a comparison, we also ran the upscaling with our method, TBP kriging, and compared our method with the three commonly used methods regarding the evaluation matrices. We included both morning and afternoon data to show the full profile of the observations.

This experimental design is anticipated to figure out the validity of TBP kriging to deal with the spatial autocorrelation detected in s dense soil moisture network.

*2.3. Spatial Autocorrelation Detection with Moran's I*

Moran's I index measures spatial autocorrelation based on both feature locations and feature values simultaneously. Given a set of features and an associated attribute, it provides a single value to evaluate whether the pattern expressed is clustered, dispersed, or random [15,18]. To detect/address where spatial autocorrelation exists in the data, this study used the local indicator of spatial association, Anselin local Moran's I [16,19]. For each observation of the data, this local Moran's I index indicates the extent of significant spatial clustering of similar values around that observation [17].

Given a pair of spatial features, either similar or dissimilar in attributes, their proximity will determine how similar they are in spatial location. In other words, spatial autocorrelation compares the two sets of similarities. If features that are similar in location also tend to be similar in attributes, then this is showing positive spatial autocorrelation; conversely, when features that are similar in location also tend to be dissimilar in attributes, then negative spatial autocorrelation is said to show [15].

Equation (1) calculates the Moran's I Index value and both z-score (standard deviation) and *p*-value (statistical significance, lower *p*-value indicate high confidence level) to evaluate the significance of that index.

$$I_i = \frac{z_i}{m_2} \sum_j w_{i,j} z_j \tag{1}$$

where $z_i$ is the deviation of the variable of interest with respect to the mean; $w_{i,j}$ is the matrix of weights that in some cases is equivalent to a binary matrix with ones in position *i,j* whenever observation *i* is a neighbor of observation j, and zero otherwise; $m_2$ is the second moment (a consistent estimate of the variance):

$$m_2 = \sum_i \frac{z_i^2}{n} \tag{2}$$

where $z_i$ is the deviation of the variable of interest with respect to the mean.

The Anselin local Moran's I Index value is unitless, but high positive values imply that the variable being measured is a cluster of high values (HH), a cluster of low values (LL), an outlier in which a high value is surrounded primarily by low values (HL), or an outlier in which a low value is surrounded primarily by high values (LH). For any of these four cases, an autocorrelation is detected. A value of 0 implies complete spatial randomness [17].

*2.4. Soil Moisture Upscaling Common Models*

Assume that the vector $\theta_{POINT}$ contains point-scale soil moisture observations sampled from a remotely sensed footprint. Defining upscaling function Fu, this set of point-scale measurements can be upscaled to represent an estimate of mean footprint-scale soil moisture.

$$\Theta_{upscale} = F_u(\Theta_{point}) \tag{3}$$

where $\Theta_{upscale}$ is the target soil moisture content at 1 km scale (large scale) and $\Theta_{point}$ is the soil moisture observation from station scale (local scale).

The commonly used upscaling models, Thiessen polygon, arithmetic average, and Gaussian-weighted average, all have their unique equations for $F_u$.

2.4.1. Thiessen Polygon

The Thiessen polygon method is selected as the default approach by the SMAP science team, which used a Voronoi diagram to find the weighting of the stations [26]. This Voronoi diagram is a partitioning of a plane into regions that are used to divide the area covered by the input point features into polygons. The polygons are named Thiessen polygons. Each Thiessen polygon contains only a single point input feature, i.e., a soil moisture station, as shown in Figure 2. Any location within a Thiessen polygon is closer to its associated station than to any other station.

The procedure to create Thiessen polygons is as follows [43]:

(1) All points are triangulated into a triangulated irregular network (TIN) that is performed on Delaunay conforming triangulations.

(2) The perpendicular bisectors for each triangle edge are generated, forming the edges of the Thiessen polygons. The location at which the bisectors intersect determines the locations of the Thiessen polygon vertices.

The Thiessen polygon method is a weighted-average method, where the area of each polygon is considered as the weight for each soil moisture station. The Thiessen polygons were created using ArcGIS.

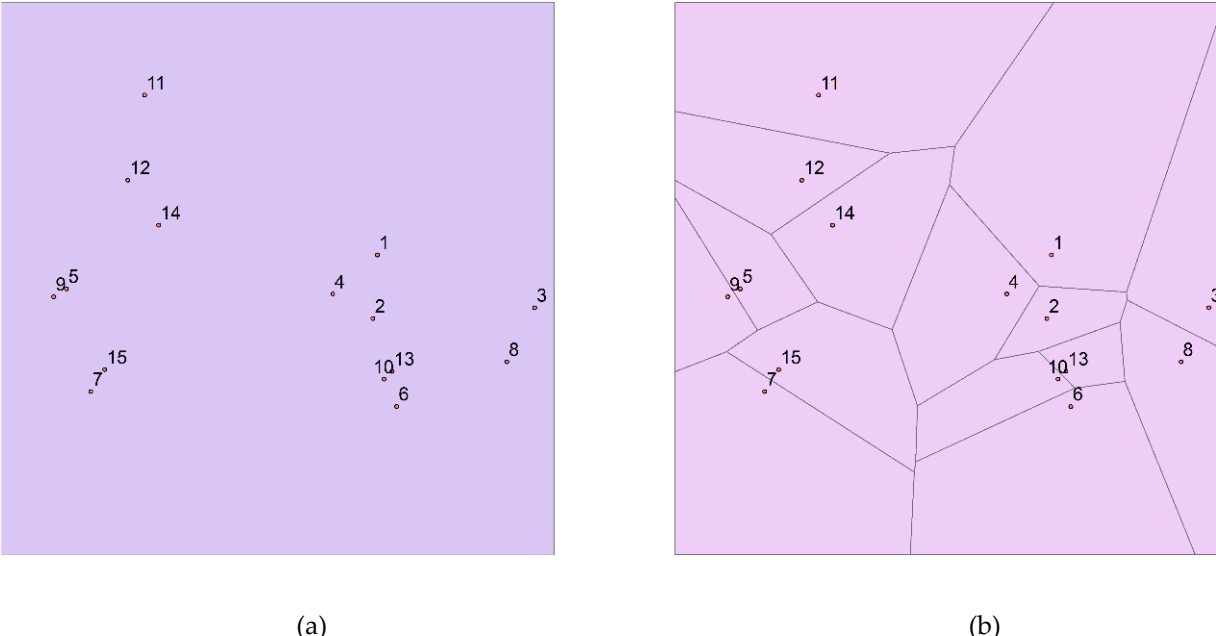

(a)　　　　　　　　　　　　　　　　　　　　　　　　　　　　　(b)

**Figure 2.** A schematic plot showing the Thiessen polygon generated from the soil moisture stations. (**a**) TxSON stations (stations 1–15) within Grid 2, the dimension of the grid is 9 km by 9 km; (**b**) Thiessen polygons generated from the stations (stations 1–15).

### 2.4.2. Arithmetic Average

The arithmetic average, or arithmetic mean, is simply the mean or average of the stations being measured. The arithmetic mean of a set of station observations is defined as being equal to the sum of the numerical values of each station observation divided by the total number of station observations. As shown in Equation (4),

$$\theta = \frac{\sum_1^{15} \theta i}{15} \tag{4}$$

where $\theta i$ is the soil moisture of each station observation.

### 2.4.3. Gaussian-Weighted Average

The SMAP soil moisture products, L3_SM_P_E, were retrieved from the brightness temperature. Thermally generated radiometric sources have an amplitude probability distribution function that is Gaussian in nature [38,44].

The spatial convolution weights were modeled as the 2D Gaussian kernel function, which is higher if the location of the station is closer to the center of the 9 km square unit and vice versa. The Gaussian kernel, the 1D Gaussian kernel, is defined as [45]:

$$Y(t) = \int K(t, s)X(s) \, ds \tag{5}$$

where $K$ is the kernel of the integral. Given the input signal $X$, $Y$ represents the output signal.

To test the performance of the Gaussian-weighted-average method, a Gaussian kernel was applied to the TxSON soil moisture product as the weight of average and compared with the SMAP 9 km grid. To implement the Gaussian-weighted average, we used the Gaussian kernel by calculating the Euclidean distance from each station within the grid

(grid 2 and 11) to the center of the 9 km grid (grid 2 and 11), then used the equation [46,47] to get the weight of each station:

$$\text{Gaussian smoothing filter} = (1/(2\pi\sigma^2))\exp(-(x^2 + y^2)/2\sigma^2) \qquad (6)$$

where $x$ and $y$ are the horizontal and vertical coordinates on a 2D map, and $\sigma$ is the standard deviation of the Gaussian distribution, which determines the width of the Gaussian kernel.

Then we used the sum of the product (combinational logic as two or more products are summed together) of the raw value and Gaussian weight to get the final value of the grid.

### 2.5. Our Method: Thiessen-Polygon-Based Block Kriging (TPB Kriging)

The following spatial model is proposed in this study to integrate the Thiessen polygon into the block kriging to handle the spatial autocorrelation in the dataset:

$$Fu = BK[TP(\Theta_{\text{point}})] \qquad (7)$$

where TP is the Thiessen polygon, BK is block kriging, and $\Theta_{\text{point}}$ is the soil moisture observation from the station scale (local scale).

To enable the capability of dealing with spatial autocorrelation, we integrated block kriging into the Thiessen polygon algorithm. Block kriging is a spatial interpolation method that predicts the average value of a phenomenon within a specified area. Block kriging provides an estimate for a discrete area around an interpolation point. A block is defined as a rectangular area or irregular block around a point that is not included in an adjacent block [9]. Block kriging performs an estimate, not for an unknown point, but for a block or area. Block kriging is used to enhance ordinary kriging, where a major disadvantage of ordinary kriging is that when sample values change very fast within short distances, ordinary point kriging may result in surfaces that have many sharp spikes or pits at the data points. Under such circumstances, block kriging acts as a method for smoothing such structures by dividing the whole area into several blocks and calculating a simple local average for each of them.

To implement TPB kriging, we applied the block kriging method using the Thiessen polygon as the input blocks and the 9 km fishnet polygon as the output blocks (Figure 3). This integration of block kriging into the Thiessen polygon tested the model definition as proposed in Equation (7), where the geometric weight (the area of each polygon) can be reserved from the Thiessen polygon, whereas the spatial autocorrelation can be measured and mitigated by the block kriging. The results were then averaged to compare with the satellite measurement. Block kriging works by calculating predictions for several specified locations within an area; the values are averaged, and the average is assigned as the prediction for the entire area [48]. The approach is implemented in ArcMap software via the geostatistical analyst tool.

The unique function provided by the block kriging is that the average expected value in an area around an unsampled point is generated rather than the estimated exact value of an unsampled point. The use of block kriging over ordinary kriging, is that in block kriging, the semi-variances between the data points and the interpolated point, which was used in kriging, are replaced by the average semi-variances between the data points and all points in the region. The literature shows that block kriging is commonly used to provide better variance estimates and smooth interpolated results than point-based ordinary kriging [9,48]. Compared with point-based ordinary kriging, block kriging can yield smaller estimation variances and smoother maps [48].

The results were then evaluated against the default Thiessen polygon method, as well as other two methods, arithmetic average and Gaussian-weighted average, for accuracy assessment.

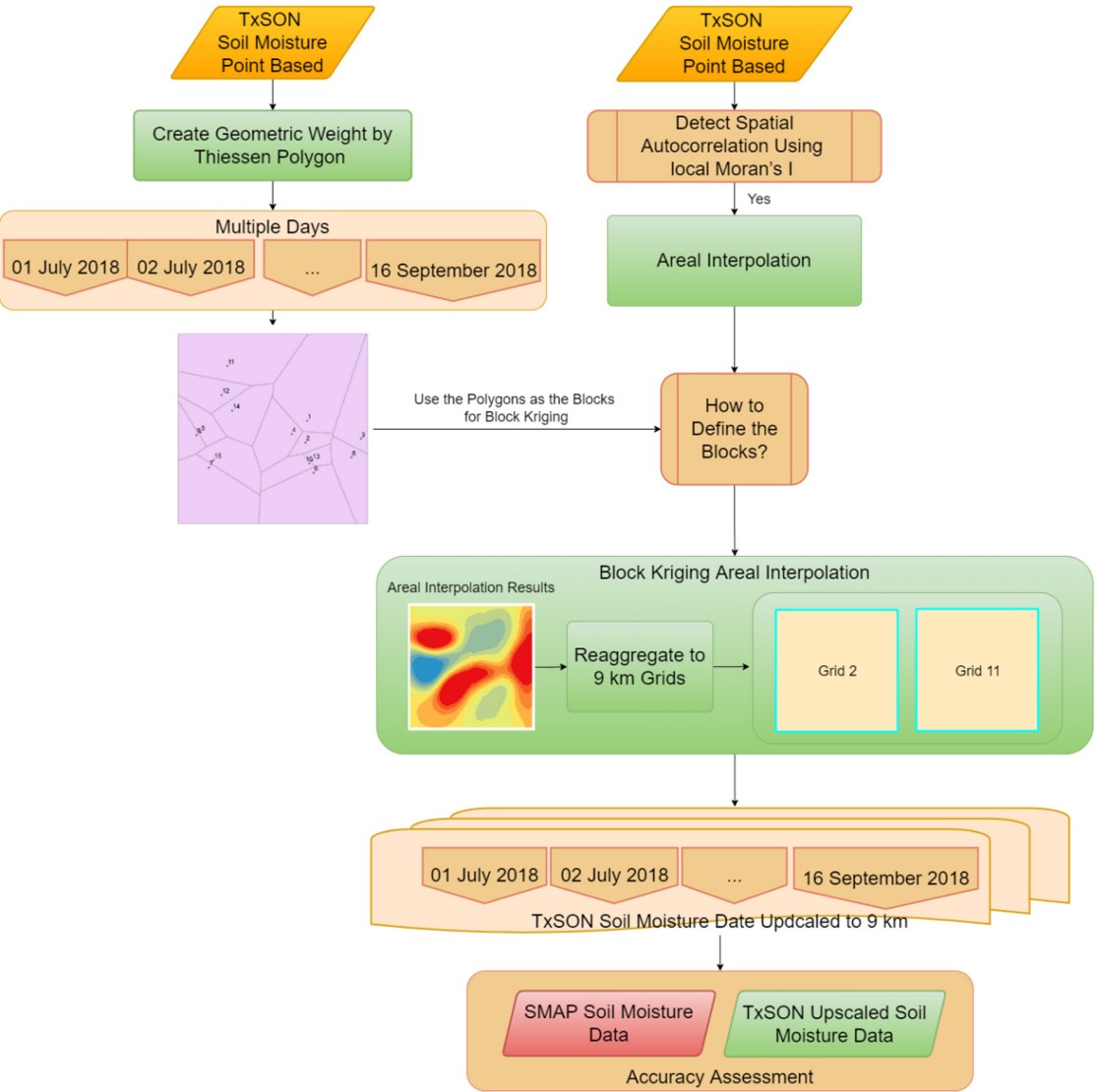

**Figure 3.** A schematic plot of the implementation of block kriging on top of the Thiessen polygon generated from the soil moisture stations (numbered 1–15). When spatial autocorrelation was detected, block kriging is used to remove the autocorrelation, using Thiessen polygons as blocks for the areal interpolation (blue indicates low soil moisture content and red indicates high soil moisture content), rather than point/pixel-based interpolation.

### 2.6. Accuracy Assessment

The accuracy was assessed based on the comparison between the SMAP 9 km grids and the upscaled results from the four algorithms. Two matrices were used for the evaluation: root-mean-square deviation (RMSD) and unbiased root-mean-square deviation (ubRMSD) [49].

$$\text{RMSD} = \sqrt{E\left[\left(\theta est - \theta true\right)^2\right]} \tag{8}$$

where $\theta_{est}$ is the estimated soil moisture from satellites and $\theta_{true}$ is the soil moisture upscaled from the ground truth data.

$$\text{ubRMSD} = \sqrt{\frac{\sum_{i=1}^{N}\left(\left(x_i - \bar{x}\right) - \left(y_i - \bar{y}\right)\right)^2}{N-1}} \tag{9}$$

where $x$ and $y$ are two sources of soil moisture observations.

The relationship between RMSD and ubRMSD is:

$$\text{RMSD}^2 = \text{ubRMSD}^2 + \text{bias}^2 \tag{10}$$

where bias is the difference between soil moisture estimate $\theta_{est}$ and the true value $\theta_{true}$.

## 3. Results

### 3.1. Spatial Autocorrelation Results from Moran's I: A Tale of Two Grids

Using Moran's I as the indicator, the two 9 km grids, i.e., grid 2 and grid 11, yielded two different scenarios of spatial autocorrelation. The Moran's I result is shown in Table 2, where grid 2 shows two stations with spatial autocorrelation (HL and LH detected), while grid 11 shows no spatial autocorrelation (no HH, LL, HL, or LH detected). Consequently, the corresponding results showed significant differences between the soil moisture upscaling results among the four algorithms, as well as their performance. To visually show the stations with the spatial autocorrelation, we produced the map of Moran's I (Figure 4). Whereas grid 2 (Figure 4a) detected spatial autocorrelation with Moran's I, grid 11 (Figure 4b) did not. Note that this result shows on July 1, 2018, 06:00 only because of the large data size. The remaining days for grid 2 reported similar spatial autocorrelation for most of the days, while grid 11 did not. To report the remaining days, we also plotted the stations with spatial autocorrelation for grid 2 for the entire 78 days (Figure A1).

**Table 2.** Soil moisture stations detected with spatial autocorrelation with Moran's I (a) detected two stations with spatial autocorrelation, LH = low–high outlier, HL = high–low outlier; (b) detected no spatial autocorrelation.

| Station ID | SMC at 6AM on 07012018 | Moran's I | Z-Score | *p*-Value | Spatial Autocorrelation |
|---|---|---|---|---|---|
| (a) Grid 2: spatial autocorrelation detectable | | | | | |
| 1 | 0.0917 | −0.5107 | −1.1894 | 0.1420 | not detectable |
| 2 | 0.1606 | −0.0637 | −0.0905 | 0.4580 | not detectable |
| 3 | 0.1631 | 0.6565 | 0.9254 | 0.2860 | not detectable |
| 4 | 0.1712 | 0.1166 | 0.3375 | 0.3740 | not detectable |
| 5 | 0.1825 | −1.4172 | −1.2253 | 0.0920 | not detectable |
| 6 | 0.0676 | 0.4762 | 0.7489 | 0.2420 | not detectable |
| 7 | 0.0568 | −2.2919 | −1.7601 | 0.0020 | LH |
| 8 | 0.1678 | 0.6565 | 0.8481 | 0.2780 | not detectable |
| 9 | 0.0598 | −1.4172 | −1.1750 | 0.0600 | not detectable |
| 10 | 0.1158 | 0.1033 | 0.9660 | 0.2120 | not detectable |
| 11 | 0.0977 | −0.2507 | −0.4384 | 0.4440 | not detectable |
| 12 | 0.1493 | −0.3991 | −1.1619 | 0.1660 | not detectable |
| 13 | 0.0838 | 0.1805 | 0.4557 | 0.4000 | not detectable |
| 14 | 0.0727 | −0.4925 | −0.5078 | 0.4060 | not detectable |
| 15 | 0.2145 | −2.2919 | −1.3332 | 0.0020 | HL |
| (b) Grid 11: spatial autocorrelation not detectable | | | | | |
| 1 | 0.0858 | −0.2477 | −0.6361 | 0.2740 | not detectable |
| 2 | 0.0998 | −0.4157 | −1.2092 | 0.0940 | not detectable |
| 3 | 0.2094 | −0.6541 | −0.7180 | 0.2440 | not detectable |
| 4 | 0.0599 | −0.6233 | −1.4407 | 0.0960 | not detectable |
| 5 | 0.1527 | 0.1737 | 1.4382 | 0.0740 | not detectable |
| 6 | 0.0606 | 0.0152 | 0.2285 | 0.4540 | not detectable |
| 8 | 0.1423 | 0.0158 | 0.2900 | 0.3620 | not detectable |
| 9 | 0.1745 | −0.5010 | −0.9394 | 0.1780 | not detectable |
| 10 | 0.0745 | −0.0837 | −0.0615 | 0.3980 | not detectable |
| 11 | 0.0834 | −0.1007 | −0.2809 | 0.3820 | not detectable |
| 12 | 0.1442 | −0.0632 | −0.1935 | 0.4600 | not detectable |
| 13 | 0.0787 | −0.1732 | −0.5079 | 0.3340 | not detectable |
| 14 | 0.1139 | 0.0008 | 0.7361 | 0.2540 | not detectable |
| 15 | 0.1126 | 0.0094 | 0.7640 | 0.2900 | not detectable |

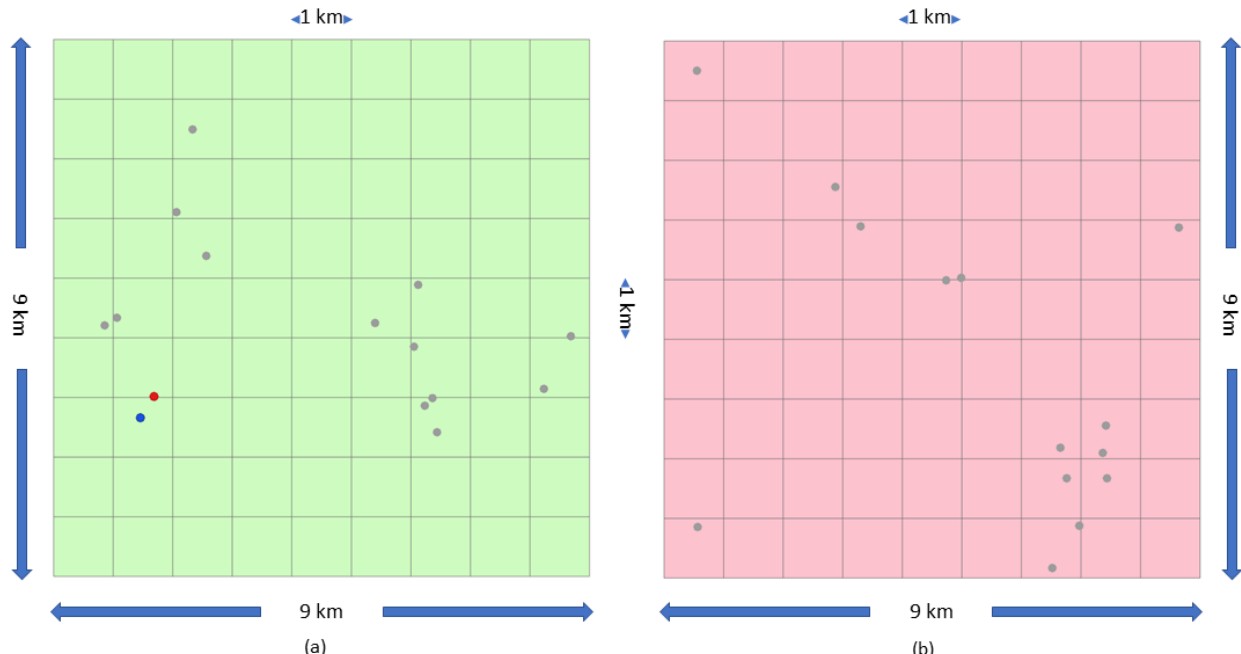

**Figure 4.** A comparison between two scenarios: spatial autocorrelation detectable versus non-detectable in the soil moisture network. (**a**) Scenario a shows spatial autocorrelation detected in grid 2, the outlier stations detected by the local Moran's I are represented in red and blue colors, where red represents a high–low (HL) outlier, blue represents a low–high (LH) outlier, and normal stations are shown in gray color; (**b**) scenario b shows spatial autocorrelation not detected in grid 11, and the normal stations are shown in gray color. Figure 4a shows an example from a single day (1 July 2018 06:00); for a completed spatial autocorrelation detected in the data, see Figure A1 in the Appendix A.

Semivariogram is another method to depict the spatial autocorrelation of the measured samples by computing and plotting the variance between any pair of soil moisture observations, and a model is then fit through them. Grid 2 is the case of detectable spatial autocorrelation. Figure 5a shows that at closer distances, the modeled soil moisture (blue line) is more predictable and has less variability in soil moisture values; in other words, the semi-variance is small. However, when the distances among the soil moisture stations are farther away, they are less predictable and the variability is increasing. This means that the semi-variance becomes large.

The model of the semivariogram shows that as the distance increases, at a certain distance there is no longer a relationship between the sample soil moisture values. The distance where the model first flattens out is known as the range. Soil moisture locations separated by distances closer than the range are spatially autocorrelated, whereas locations farther apart than the range are not spatially autocorrelated. Figure 5a shows that when the distance (x-axis) increases from 727 m to 5091 m, the modeled soil moisture semivariance (blue line) is increasing; the semivariance continues increasing from 5091 m to 5818 m, then flattens out afterward. This is a perfect example of spatial autocorrelation.

However, in the case of grid 11, a different scenario is observed from the results. Figure 5b shows that the semivariogram and the model are very different from Figure 5a in that the semivariance in the y-axis does not increase with distance as significant as (a). The model (blue line) flattens out from the very beginning, somewhere between 727 m and 1455 m. The relatively stable semivariance indicates that the autocorrelation is not detected in the soil moisture data with grid 11.

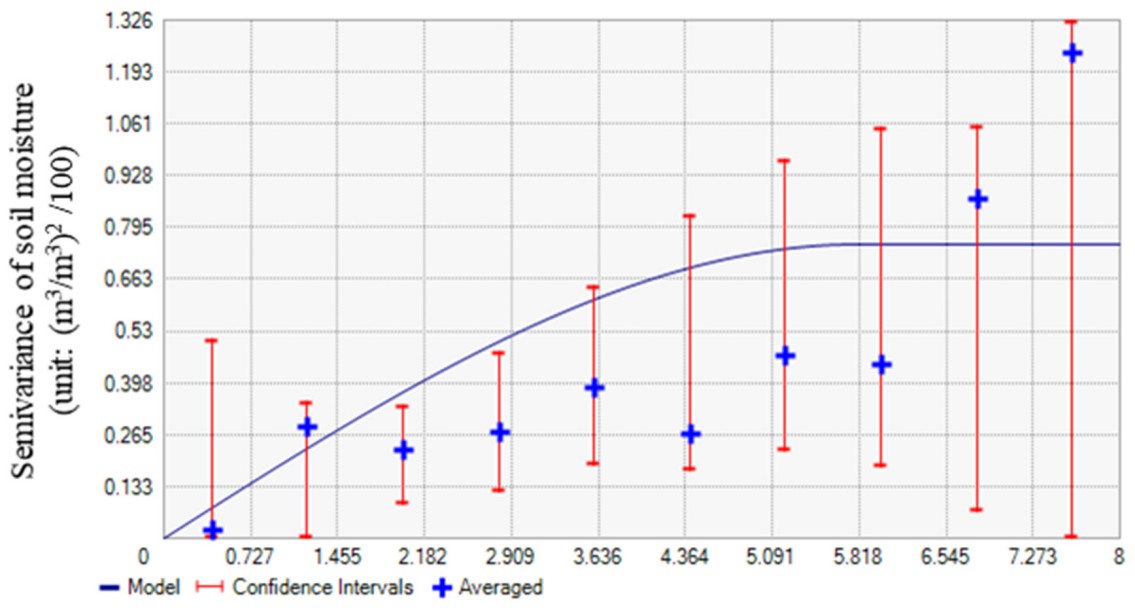

Model: 0.007523*Spherical(5794.7)

(a)

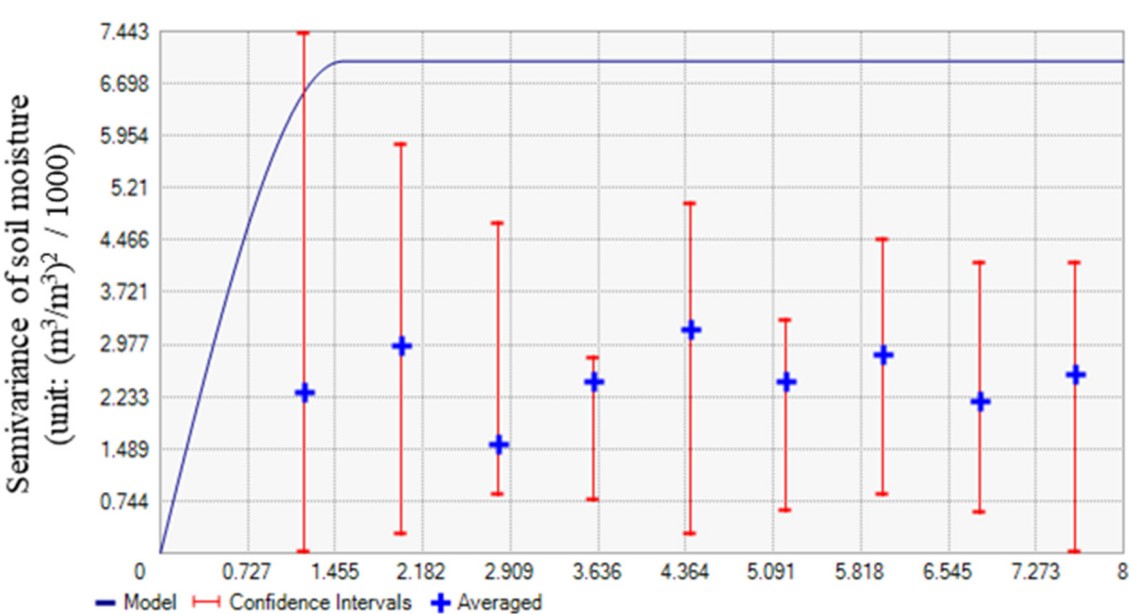

Model: 0.0070154*Spherical(1515.5)

(b)

**Figure 5.** Semivariograms from the block kriging for the spatial autocorrelation. (**a**) Scenario a shows spatial autocorrelation detected in grid 2; (**b**) scenario b shows spatial autocorrelation not detected in grid 11. The semivariogram depicts the spatial autocorrelation of the measured sample soil moisture stations (points). A model (blue line) is fit through the average of each pair of locations (blue-colored crosses). The semivariograms for the two scenarios were modeled by spherical equations, which are the mathematical description of the two blue lines.

*3.2. Scenario One—When Spatial Autocorrelation Is Detected in the Data*

3.2.1. Daily Soil Moisture Trend from the Four Upscaling Algorithms Compared with SMAP

The results include the daily trend analysis for SMAP and different upscaling methods. The daily trend was based on soil moisture and precipitation time-series data from 1 July to 16 September 2018. The correlation plots between SMAP and each upscaling method, as well as the matrix table, were reported for performance assessment.

Figure 6 shows the daily trend analysis in the morning based on both SMAP and the results of arithmetic average, Thiessen polygon, Gaussian-weighted average, and TBP kriging. The daily soil moisture trend in the morning shows that the soil moisture from 1 July to 5 July (SAMP) or 8 July (TxSON) has an increasing trend, followed by a decreasing trend until 8 August. Another increase was observed from 8 August to 12 August (TxSON) or 14 (SMAP), and generally a decrease until September 1. The trend thereafter had several ups and downs. From 1 July to 16 September, three soil moisture peaks occurred: 5 or 8 July, 12 or 14 August, and 15 September. The pikes shown in the soil moisture generally correlated with the days that precipitation was observed (Figure A2).

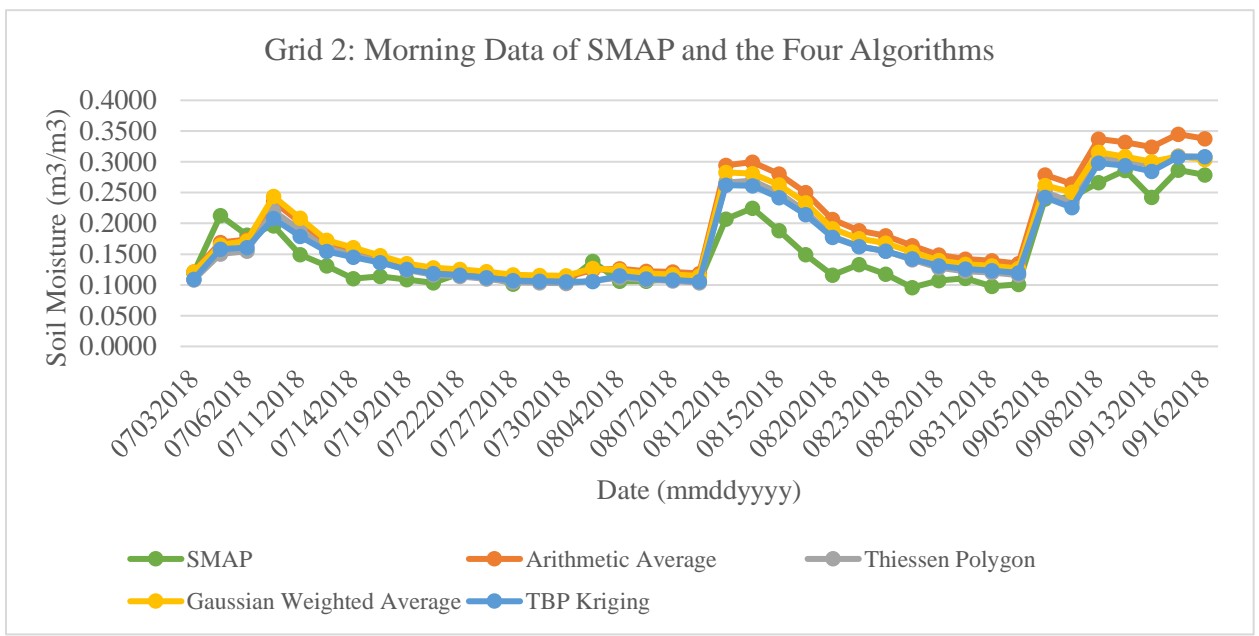

**Figure 6.** Grid 2—spatial autocorrelation detected: morning soil moisture data plot with SMAP and the four algorithms. Soil moisture was measured at 5 cm depth. Orange line: time plot of SMAP and arithmetic average for the morning data for the TxSON stations from June to September 2018. Gray line: time plot of SMAP and Thiessen polygon method for the morning data for the TxSON stations from June to September 2018. Yellow line: time plot of SMAP and Gaussian-weighted average for the morning data for the TxSON stations from June to September 2018. Blue line: time plot of SMAP and TBP kriging for the morning data for the TxSON stations from June to September 2018.

Figure 7 shows the daily trend analysis in the afternoon based on both SMAP and the results of arithmetic average, Thiessen polygon, Gaussian-weighted average, and TBP kriging. The daily soil moisture trend in the afternoon shows that from 1 July to 5 July (SAMP) or 7 July (TxSON) has an increasing trend, followed by a decreasing trend until 8 August. Another increase was observed from 8 August to 13 August (TxSON) or 14 (SMAP), and generally a decrease until 1 September. The trend thereafter had several ups and downs. From July 1 to 16 September, three soil moisture peaks occurred: 5 or 7 July, 13 or 14 August, and 15 September. The pikes shown in the soil moisture generally correlated with the days that precipitation was observed (Figure A2).

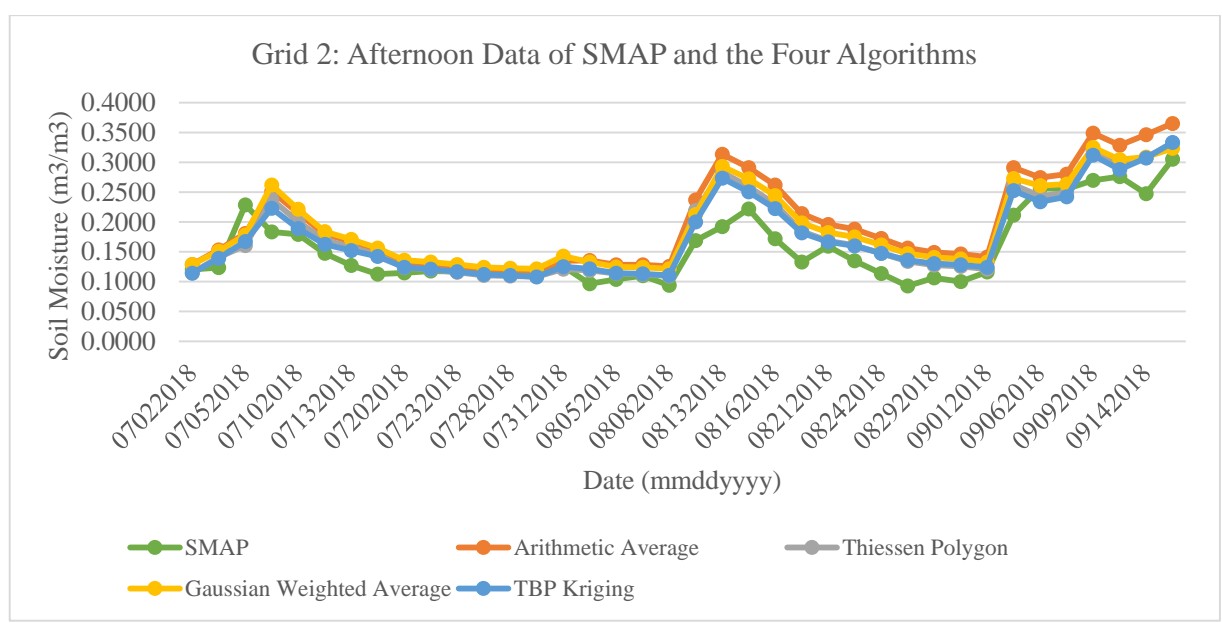

**Figure 7.** Grid 2—spatial autocorrelation detected: afternoon soil moisture data plot with SMAP and the four algorithms. Soil moisture was measured at 5 cm depth.

### 3.2.2. Evaluation of the Four Upscaling Algorithms for Grid 2

The correlation plot of SMAP and the four methods for both the morning and afternoon data are shown in Figures 8 and 9. Table 3 shows the model performance comparison for both morning and afternoon data in terms of RMSD and ubRMSD.

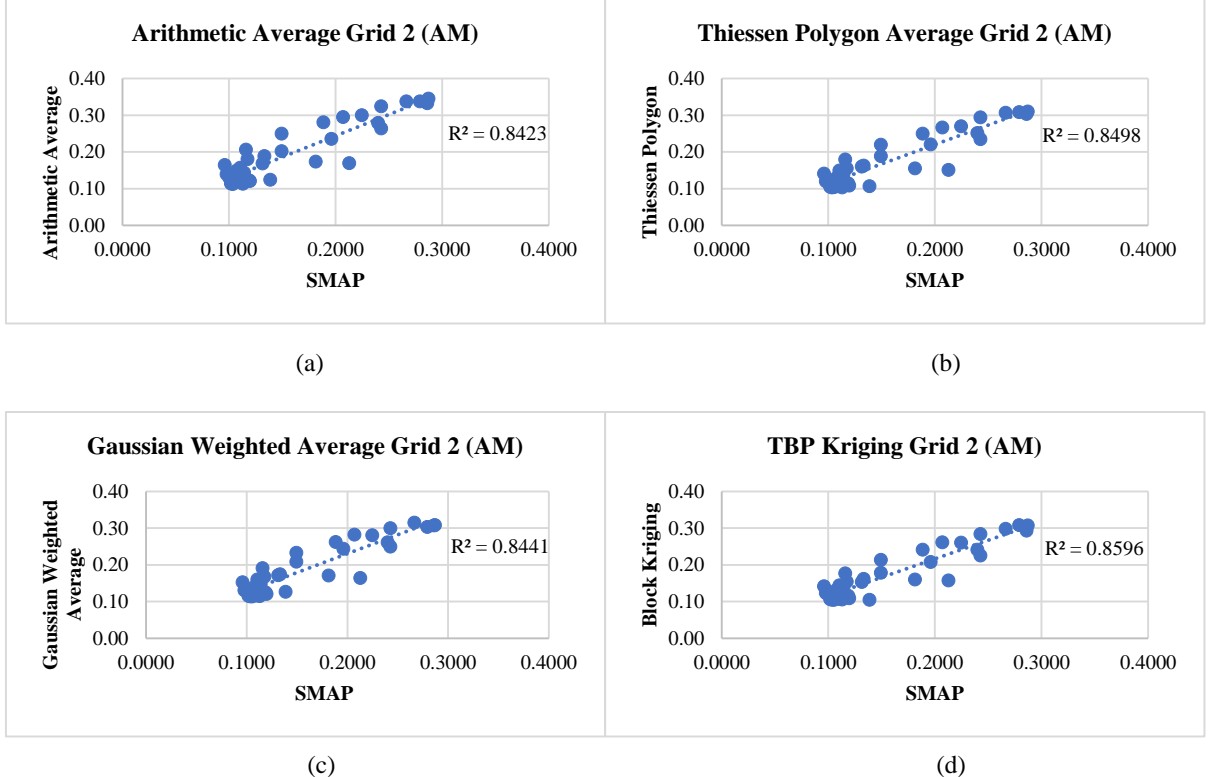

**Figure 8.** Grid 2—spatial autocorrelation detected: correlation plot between the morning SMAP data and the four upscaling algorithms. (**a**) arithmetic average; (**b**) Thiessen polygon method; (**c**) Gaussian-weighted average; (**d**) TBP kriging.

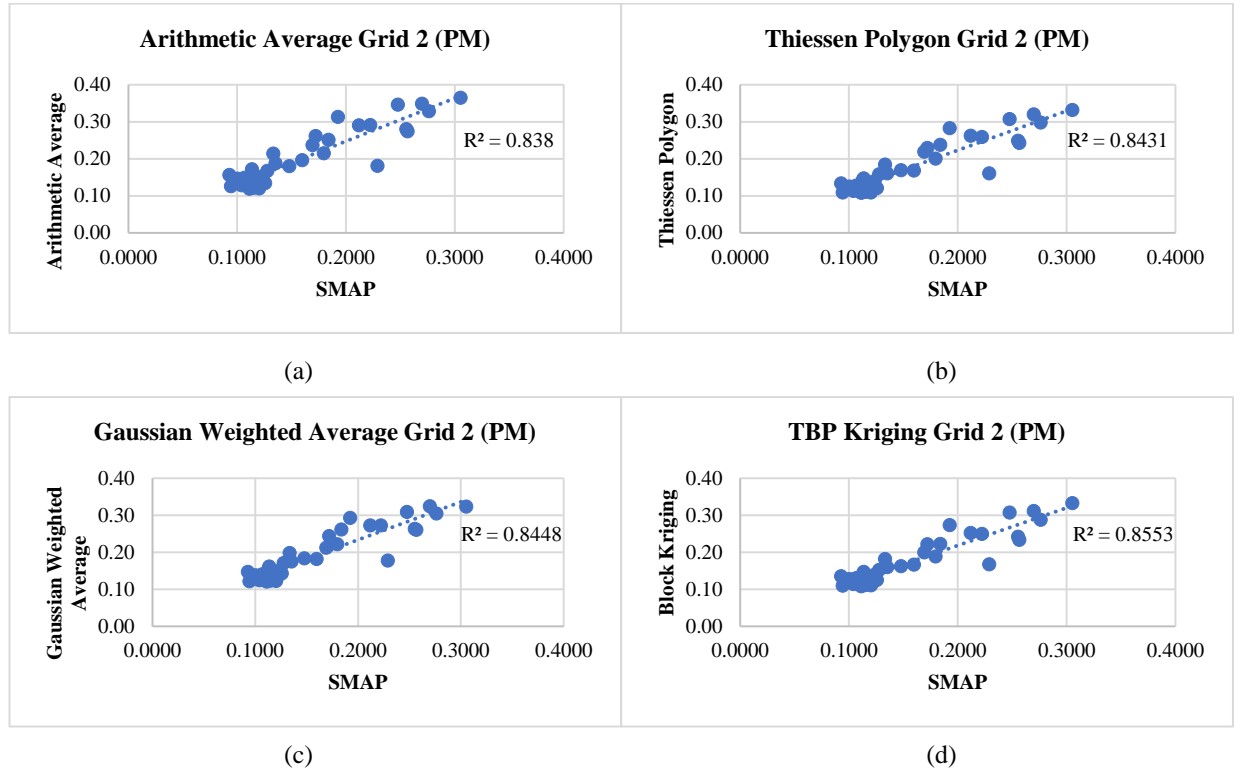

**Figure 9.** Grid 2—spatial autocorrelation detected: correlation plot between the afternoon SMAP data and the four upscaling algorithms. (**a**) arithmetic average; (**b**) Thiessen polygon method; (**c**) Gaussian-weighted average; (**d**) TBP kriging.

**Table 3.** Model Performance Comparison for the AM/PM Data for Grid 2.

| Validation Matrix | TBP Kriging | Gaussian | Thiessen Polygon | Arithmetic Average | Validation Matrix | TBP Kriging | Gaussian | Thiessen Polygon | Arithmetic Average |
|---|---|---|---|---|---|---|---|---|---|
| | Morning | | | | | Afternoon | | | |
| RMSD | 0.0298 | 0.0400 | 0.0326 | 0.0489 | RMSD | 0.0306 | 0.0419 | 0.0344 | 0.0518 |
| ubRMSD | 0.0251 | 0.0272 | 0.0275 | 0.0322 | ubRMSD | 0.0249 | 0.0262 | 0.0276 | 0.0320 |
| bias | −0.0160 | −0.0293 | −0.0176 | −0.0367 | bias | −0.0178 | −0.0326 | −0.0206 | −0.0408 |

As shown in Table 3, for both morning and afternoon data, TBP kriging yielded the best accuracy. For the morning data, the best RMSD is 0.0298 (TBP kriging method), which is smaller than all the other algorithms (0.0400, 0.0326, and 0.0489, respectively). The ubRMSD is 0.0251, which indicates there is not much bias between the two datasets. For the afternoon data, the best RMSD is 0.0306 (TBP kriging method), which is also much smaller than all the other algorithms (0.0419 of Gaussian, 0.0344 of Thiessen polygon, and 0.0518 of arithmetic average). The ubRMSD is 0.0249, which indicates there is not much bias between the two datasets. Gaussian, by comparison, yielded the least bias; however, their RMSD and ubRMSD are larger than RMSD. Comparing the morning data with afternoon data, it can be found that the RMSD is slightly larger in the afternoon than in the morning (e.g., 0.0306 and 0.0298 for BK), which is consistent with Chan et al. [39]. On the other hand, the ubRMSD in the afternoon is smaller than in the morning, which means the variance in the afternoon is smaller than in the morning, while the bias is much larger.

### 3.3. Scenario Two—When Spatial Autocorrelation Is Not Detected in the Data

3.3.1. Daily Soil Moisture Trend from the Four Upscaling Algorithms Compared with SMAP

Figures 10 and 11 show the daily trend analysis based on both SMAP and the results of arithmetic average, Thiessen polygon, Gaussian-weighted average, and TBP kriging

for grid 11. The pikes shown in the soil moisture generally correlated with the days that precipitation was observed (Figure A2). The daily soil moisture trend in the morning (Figure 10) and afternoon (Figure 11) both show a small variation among the four upscaling algorithms, compared with grid 2 (Figures 6 and 7). The nuance between grid 2 and grid 11 is consistent with the hypothesis of this study: when spatial autocorrelation is detected, as represented by grid 2, only the TBP kriging method accounted for the spatial autocorrelation in the data, yielding a soil moisture time plot very close to that of SMAP (the blue lines as shown in Figures 6 and 7). On the contrary, in the case of grid 11, the four algorithms performed very similarly as a result of spatial autocorrelation not existing in the data.

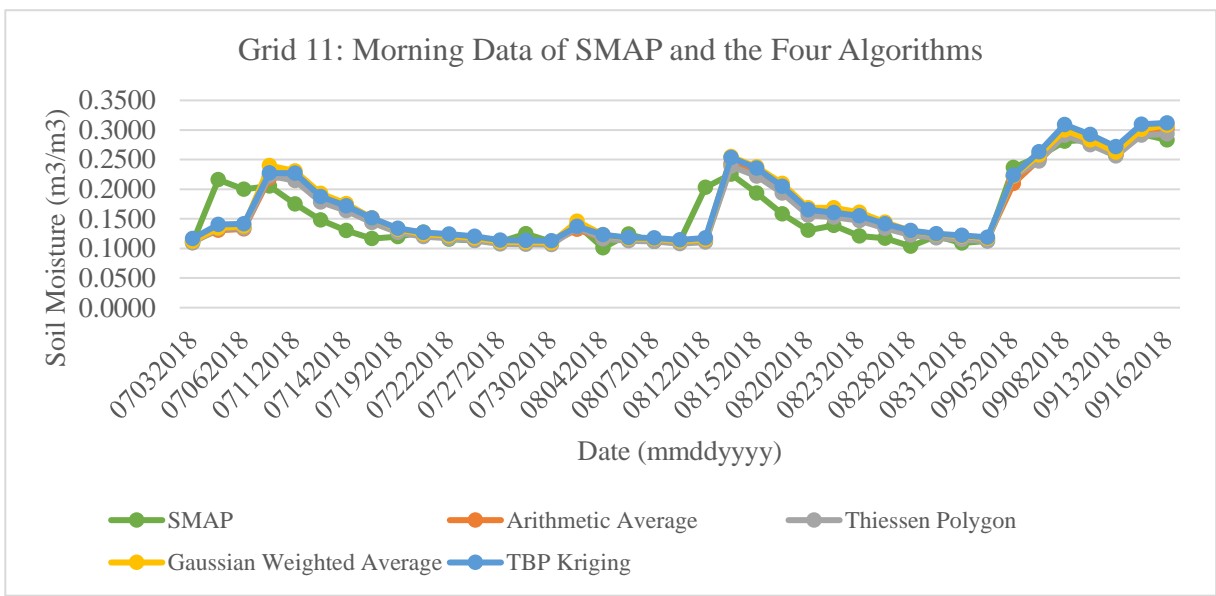

**Figure 10.** Grid 11—spatial autocorrelation not detected: morning soil moisture data plot with SMAP and the four algorithms. Soil moisture was measured at 5 cm depth.

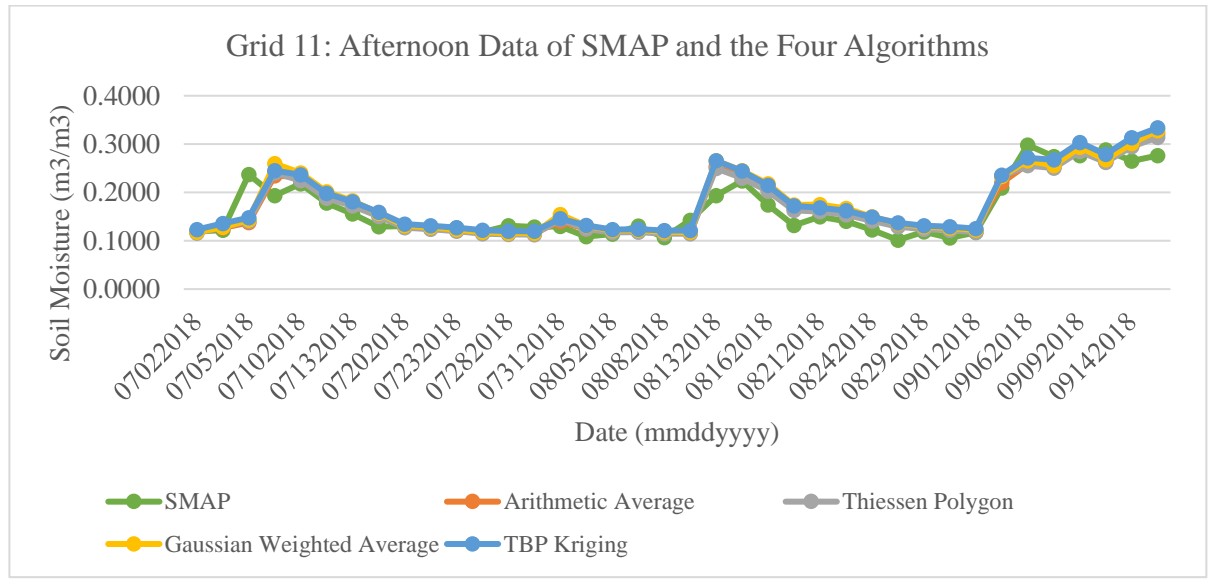

**Figure 11.** Grid 11—spatial autocorrelation not detected: afternoon soil moisture data plot with SMAP and the four algorithms. Soil moisture was measured at 5 cm depth.

### 3.3.2. Evaluation of the Four Algorithms for Grid 11

The evaluation from the four algorithms for grid 11 revealed a different story, where the Thiessen polygon outperforms the other three algorithms, including TBP kriging. Figure 12 and Table 4 show that the Thiessen Polygon method yielded a lower RMSD and ubRMSD, with higher R-squared compared with the other three algorithms; Figure 13 shows that TBP kriging achieved a higher R-squared; however, both RMSD and ubRMSD show lower accuracy compared with the Thiessen Polygon method. As supported by the results, block-kriging methods provide a more sophisticated approach to aggregating these observations based on the observed auto-correlation structure [8].

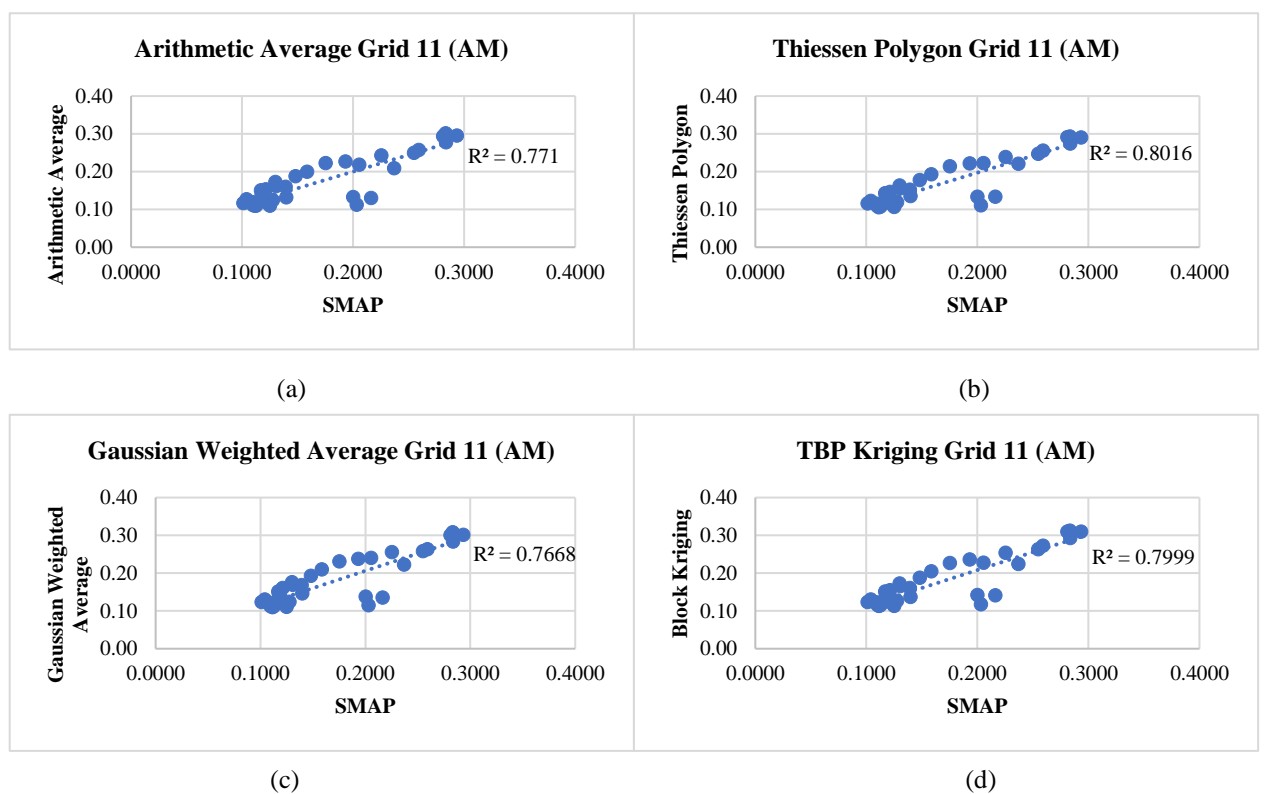

**Figure 12.** Grid 11—spatial autocorrelation not detected: correlation plot between the morning SMAP data and the four upscaling algorithms. (**a**) arithmetic average; (**b**) Thiessen polygon method; (**c**) Gaussian-weighted average; (**d**) TBP kriging.

**Table 4.** Model Performance Comparison for the AM/PM Data Grid 11.

| Validation Matrix | TBP Kriging | Gaussian | Thiessen Polygon | Arithmetic Average | Validation Matrix | TBP Kriging | Gaussian | Thiessen Polygon | Arithmetic Average |
|---|---|---|---|---|---|---|---|---|---|
| | | Morning | | | | | Afternoon | | |
| RMSD | 0.0313 | 0.0335 | 0.0289 | 0.0314 | RMSD | 0.0304 | 0.0316 | 0.0268 | 0.0282 |
| ubRMSD | 0.0297 | 0.0322 | 0.0288 | 0.0311 | ubRMSD | 0.0269 | 0.0291 | 0.0263 | 0.0271 |
| bias | −0.0100 | −0.0093 | −0.0011 | −0.0043 | bias | −0.0141 | −0.0122 | −0.0045 | −0.0078 |

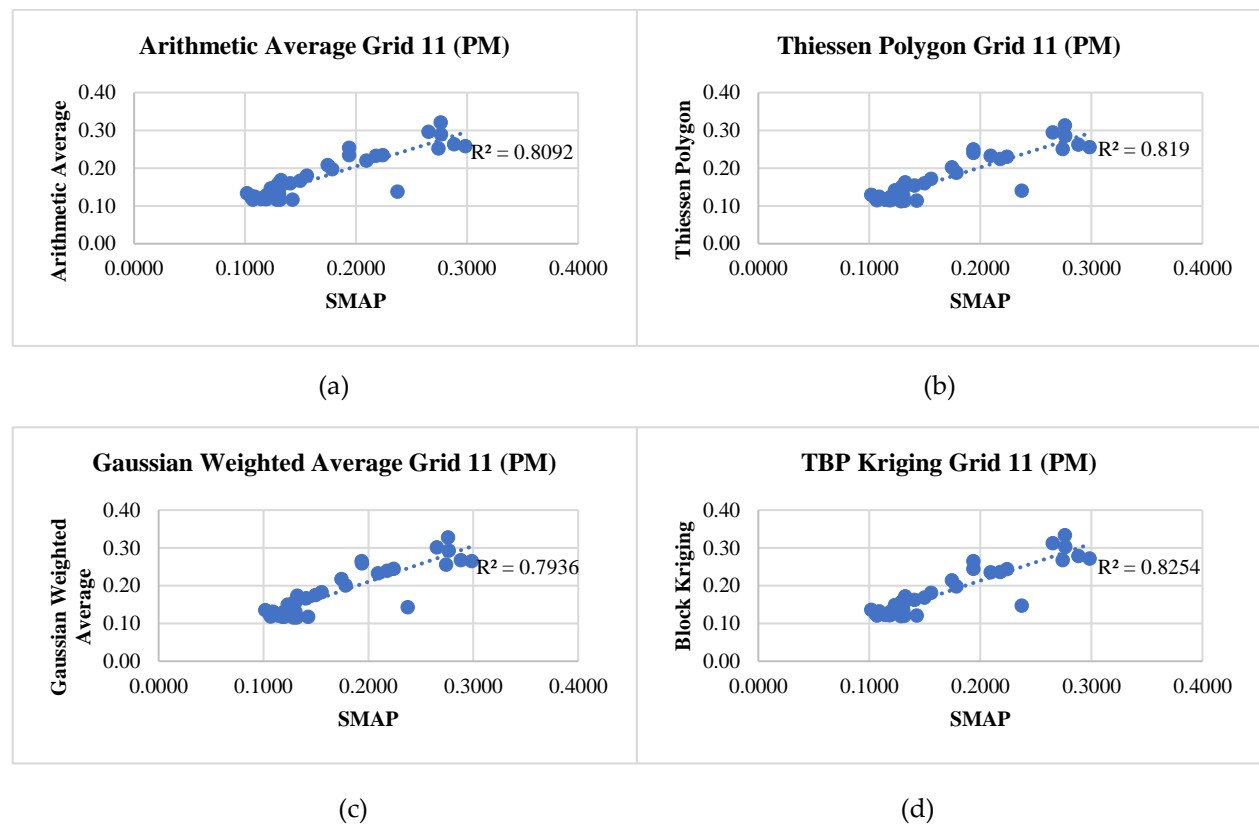

**Figure 13.** Grid 11—spatial autocorrelation not detected: correlation plot between the afternoon SMAP data and the four upscaling algorithms. (**a**) arithmetic average; (**b**) Thiessen polygon method; (**c**) Gaussian-weighted average; (**d**) TBP kriging.

## 4. Discussions

Spatio-temporal modeling of soil moisture is vital for global and regional climate change studies. The spatio-temporal pattern analysis of soil moisture is very important for flooding prediction or crop yield estimation. This research focused on the representation of spatio-temporal patterns of soil moisture based on multiscale and multisource data. Four methods were compared for the TxSON soil moisture upscaling: arithmetic average, Thiessen polygon, Gaussian-weighted average, and TBP kriging. Our method, TBP kriging, is the best overall method when spatial autocorrelation is detected, which gives the best RMSD and ubRMSD. Several topics, including when and why our method will outperform other upscaling methods, as well as its implications, are discussed below.

### 4.1. When Our Method Outperforms the Commonly Used Algorithms

The two 9 km grids, i.e., grid 2 and grid 11, are representatives of two different scenarios. When spatial autocorrelation is detected in the data, as represented by grid 2, the spatial autocorrelation can impact the upscaling model performance. Commonly used algorithms, including arithmetic average, Thiessen polygon method, and Gaussian-weighted average, underperform our method because of the absence of taking spatial autocorrelation into consideration.

### 4.2. Why Our Method Outperforms the Commonly Used Algorithms

The arithmetic average is outperformed by all three other methods, mainly because the arithmetic average does not consider the spatial weight of the contribution from each soil moisture station. The Gaussian-weighted average slightly improves the accuracy as a result of using the Gaussian kernel function to simulate the Gaussian probability distribution of the brightness temperature that generated the SMAP soil moisture products.

The Thiessen polygon method is a spatially weighted-average method, where the spatial weight for each soil moisture station is defined by the area of each polygon. However, when the stations show a "cluster" spatial pattern, the Thiessen polygons overlook the spatial autocorrelations, which significantly decreases the model accuracy. Therefore, under such circumstances, the block kriging, when working together with the Thiessen polygon, will detect and model the spatial autocorrelations while taking the area of each polygon as the spatial weight. Therefore, the TBP kriging outperforms the Thiessen polygon upscaling method and is overall the best among the four models.

Grid 11 reveals a significantly different scenario as compared to grid 2. Associating these two scenarios with the theory/calculation section, grid 2 detected spatial autocorrelation in the soil moisture network; therefore, using the proposed method, i.e., TBP kriging, was able to reduce the spatial autocorrelation of the datasets, and therefore achieve better performance in the soil moisture upscaling. On the contrary, the data from grid 11 did not detect spatial autocorrelation. In this case, the commonly used algorithm, the Thiessen Polygon, was more suitable for this type of soil moisture station distribution.

### 4.3. Implications for Soil Moisture Network Optimization

Grids 2 and 11 represent two types of soil moisture station designs. The major difference between the two is the spatial distribution of the stations. Whereas spatial autocorrelation is detected in grid 2, the clustering effect is significant, therefore causing a larger bias when compared with satellite soil moisture products. When designing a soil moisture network, we need to take into consideration the bias associated with this design. By comparison, grid 11 did not detect obvious spatial autocorrelation, which indicates a better design for soil moisture ground stations.

### 5. Conclusions

The study found that the distribution of the dataset may decrease the reliability of the dense network, as the spatial autocorrelation in the dataset might impact the model accuracy and increase the bias when comparing the network with satellite products. The slightly better performance of integrating the Thiessen polygon into the block kriging concludes that the average function within the block kriging can smooth the result using the block of 9 km.

The study also suggested that the morning soil moisture observation from SMAP performs slightly better than the afternoon soil moisture data in terms of bias, which supports the general assumptions in the literature that the soil moisture data measured in the morning is more accurate than in the afternoon from the radar or radiometer.

Overall, this study proposed a new approach for intercomparing soil moisture products from upscaling the station products and satellite products while taking the spatial autocorrelation in the soil moisture network into consideration. The discoveries of the study can potentially benefit soil moisture network design. Questions remain on the impacts of such spatial autocorrelation at a larger spatial coverage, as well as their temporal patterns, which will be addressed in future research.

**Author Contributions:** Conceptualization, Y.X. and L.W.; methodology, Y.X.; software, Y.X.; validation, Y.X. and C.L.; formal analysis, Y.X. and C.L.; investigation: Y.X.; resources, Y.X. and L.W.; data curation, Y.X. and C.L.; writing—original draft preparation, Y.X.; writing—review and editing, L.W., L.Z. and C.L.; visualization, Y.X.; supervision, L.W.; project administration: Y.X. and C.L.; funding acquisition, Y.X. and C.L. All authors have read and agreed to the published version of the manuscript.

**Funding:** The work was supported by the Louisiana State University Dissertation Fellowship. A Louisiana Sea Grant, as well as the Southern Regional Climate Center, also supported the workshops and seminars that provided opportunities for presenting and improving this work.

**Data Availability Statement:** Data availability upon request to the corresponding author.

**Acknowledgments:** The authors would like to thank Todd Caldwell, former Principal Investigator of the TxSON program and former Research Associate at the University of Texas at Austin, for collecting and providing the raw TxSON data. Thank Louisiana Sea Grant and the Southern Regional Climate Center for supporting the workshops and seminars that provided opportunities for presenting and improving this work.

**Conflicts of Interest:** The authors declare no conflict of interest. The funders had no role in the design of the study, in the collection, analyses, or interpretation of data, in the writing of the manuscript, or in the decision to publish the results.

## Appendix A

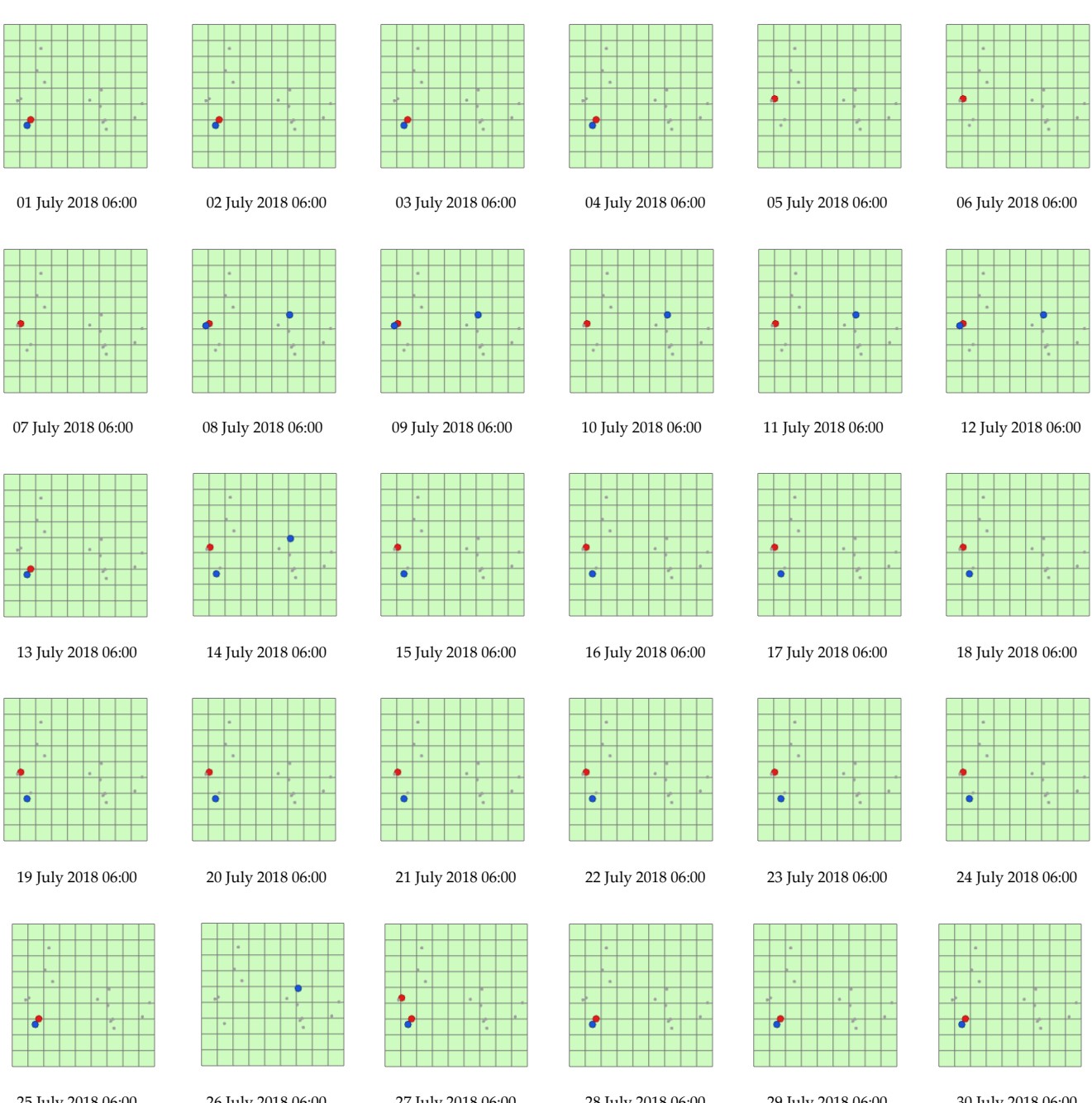

**Figure A1.** *Cont.*

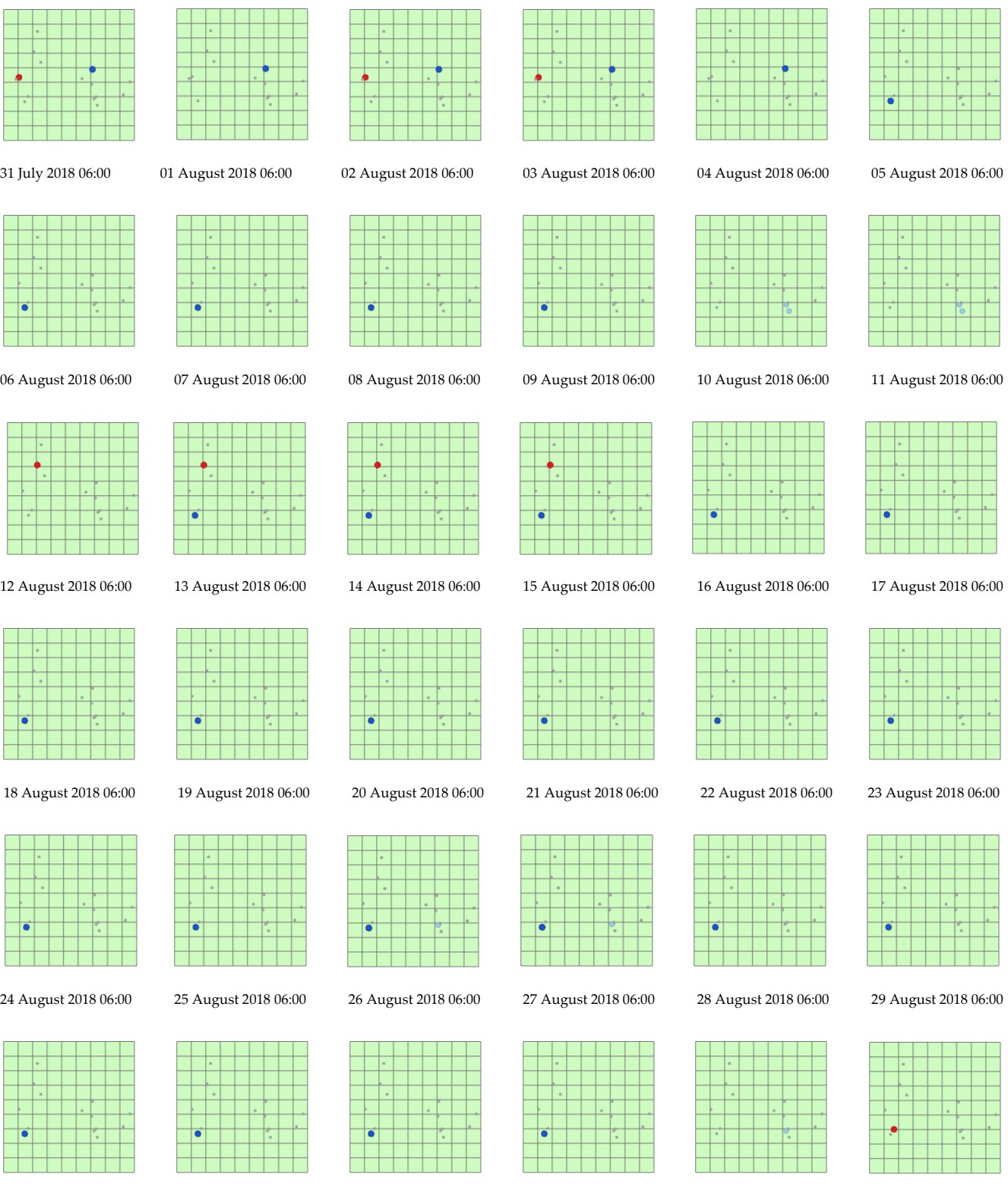

**Figure A1.** *Cont.*

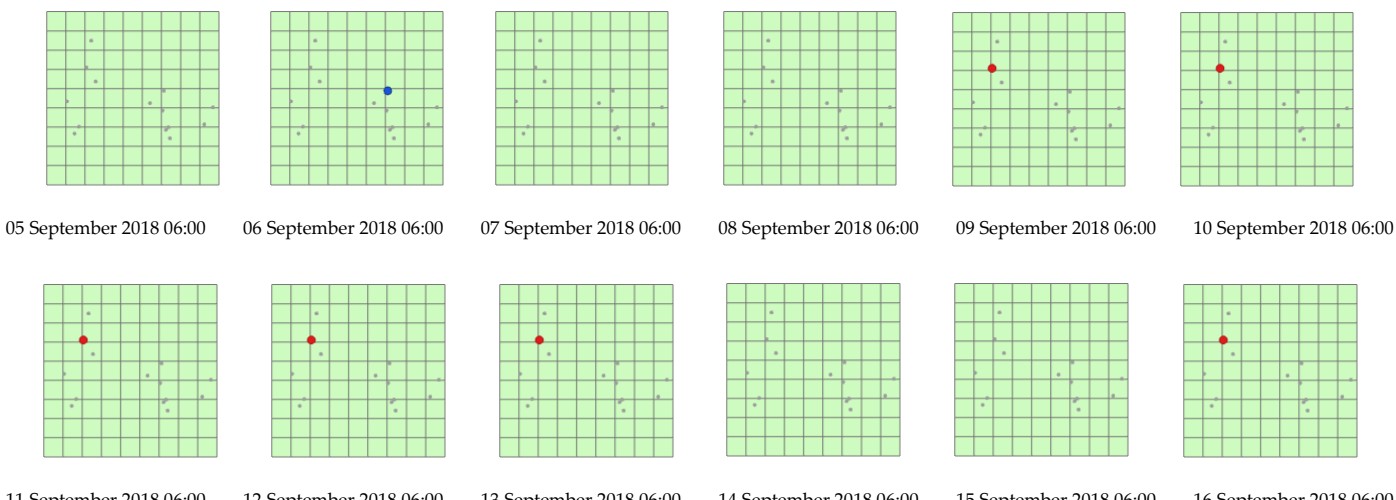

**Figure A1.** Spatial autocorrelation exists in grid 2. Red points represent high values surrounded primarily by low values (high–low outliers), and blue points represent low values surrounded primarily by high values (low–high outliers), which are two typical cluster types of spatial autocorrelation.

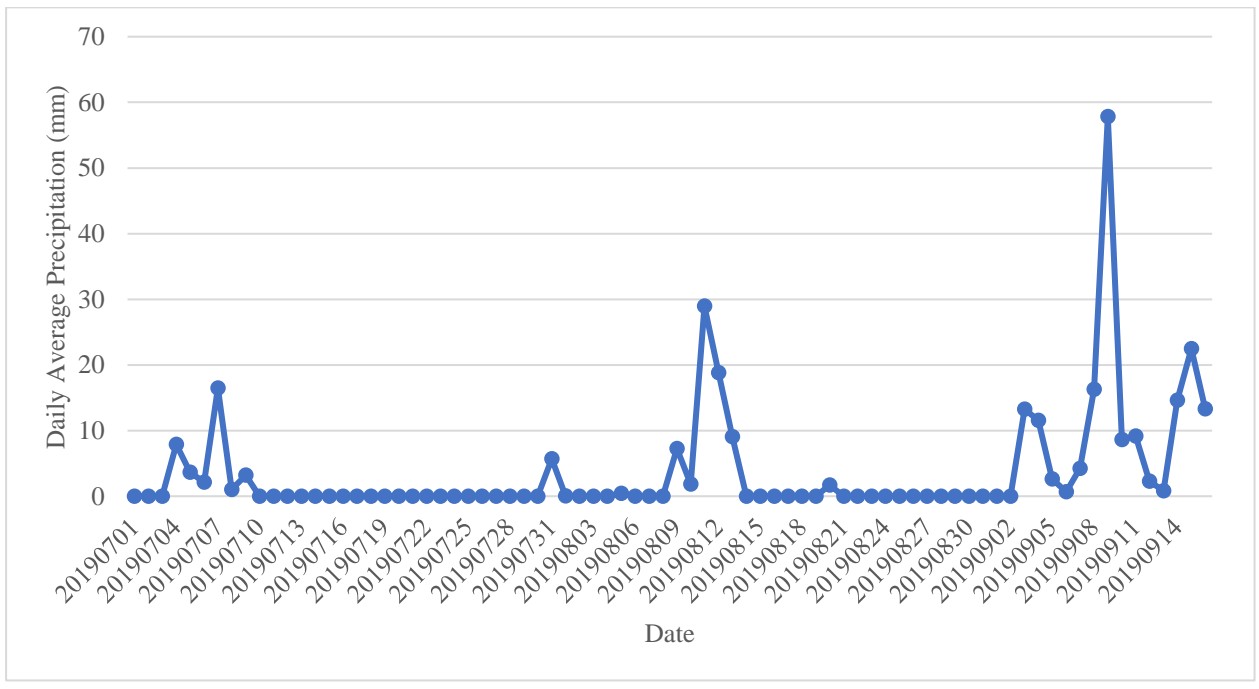

**Figure A2.** Daily average precipitation in the study area.

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
