# Peer review of "Exploring the Spatial Autocorrelation in Soil Moisture Networks: Analysis of the Bias from Upscaling the Texas Soil Observation Network (TxSON)"

_water, doi:10.3390/w15010087_

Round 1

Reviewer 1 Report

This study proposed a spatial analysis approach for soil moisture ground observation upscaling, Thiessen polygon-based block kriging, and compared the result with three other methods typically used in the current literature

It looks the authors seem to consider SMAP SWC ( soil water content ) as target.

Considering many papers tried to evaluate SMAP SWC using ground-based in situ SWC, it is a little bit confusing. it's better to make it clearer.

Also the spatial autocorrelation is explored in this study. Detailed information about the methodology and its application is necessary. And the judgement of the existence of the spatial autocorrelation needs to be clearly depicted. 

And the judgement seems to be made day by day. However, proposed method is evaluated over a specifi period, how you take into account of the time-varying autocorrelation in your study? Also how can you utilize these results to observation network design? 

If possible, also please show the impacts of the  spatial autocorrelation of rainfall.

The results show the existence of the spatial autocorrelation do make some differences in RMSD and ubRMSD. Could you show the statistical significance of these differnces?

Some minor issues:

1. Line 129 and 587:

    more accurate --> more stable

    The morning and afternoon satellate SWC are usually different.  Oanh and Lu, 2022, https://doi.org/10.3390/rs13204104) showed this is caused by temperature effects which are apparent. Same effects also exist in in situ SWC dielectrically measured (e. g. Kapilaratne & Lu, 2017, https://doi.org/10.1016/j.jhydrol.2017.05.050).  Morning SWC is relatively stable because nighttime or morning soil temperature is relatively stable.

2. Line 143

      5.08cm

       Is this exact? If so, please add refence here.

3.  Line 156

       Please add information about sensor insertion angle and if the cite calibrations are made or not.

4.  Line 157

       5-minute

       Please show local standard time and the difference between the local time and local standard time, and possibility to pick up nearest data.

5.  Line 214 and 232

     Eqs are not numbered from (1).

     Some symbols are in normal and bold face. Are they same?

     What's z-score and p-value? How they are calculated?

     More details are necessary to understand.

     Lines below eqs are all indented. Is this correct?

     From Figure A1, it looks the eq(1) is applied day by day. Please make this clear.

     If possible, please show the details about the results of eq(1) and the criteria to make judgement using Figure 4 

     Please specify date of Figure 4.

6. Figure 5

     Please make the both subfigure have same x-axis and y-axis for easier comparison.

     Some symbols are not explained.

     More informations are needed.

7. Line 340

      Please explain bias

8.  Table 2 and 3

      Please add bias. Personally I think bias is more important.

9.  misc. issues:

      in situ or in-situ    Please use either.

      line 21 Arithmetic Average  --> arithmetic average

      line 98  IDW     --> spellout

      Line 133 and 137 SMAP     define SMAP at its first appearence.

Reviewer 2 Report

In this manuscript, the authors present a new upscaling method: Thiessen polygon-based block kriging. The method is used to upscale soil moisture measurements from the Texas Soil Observation Network (TxSON). The upscaled values are then compared with the SMAP estimates of soil moisture. Overall, while the method sounds interesting, there are some major issues as highlighted below (in order of importance), which warrant a major revision:

1. The authors use SMAP soil moisture estimates as the ground truth. This assumption needs more justification and may not be a reasonable default. In Colliander et al. (2017), a key study cited by the authors, the SMAP soil moisture product is validated against upscaled soil measurements from the TxSON network. In that study, the TxSON measurements are upscaled using the Thiessen polygon method which then serve as a ground truth to evaluate SMAP estimates. The objective of that study was not to determine an optimal upscaling approach, but rather to evaluate if SMAP estimates are broadly consistent with in-situ observations. I understand that the authors’ proposed method may yield a closer match between SMAP and upscaled values, but how do we know that this is a consequence of the upscaled values being more correct? The upscaled values could simply be yielding a higher bias which could be consistent with the bias in SMAP measurements. For instance, Colliander et al. (2017) state that SMAP tends to underestimate soil moisture throughout most of the period. We observe this underestimation in all the time plots presented in the current study (Figures 6,7,10,11).

2. Section 4, Discussion, lines 527-529: What do you mean by spatial modeling? Does that refer to upscaling? If so, the statement that “spatial modeling provides a more accurate measurement than satellite-based measurement” is at odds with your approach of using SMAP as a ground truth. If satellite measurements are less accurate, then how do you know that your upscaling strategy is better?

3. The description of various methods used in the study can benefit from more details. For instance, can the authors provide more details on the equations underlying block kriging and Thiessen polygon-based block kriging? While I understand the general idea (that you are using Thiessen polygon as a block), the implementation is not clear. Given that this is the main contribution of the study, a clear description of the method is needed. 

Additionally, how did you implement the gaussian weighted average method? Is there a discrete form of the gaussian kernel that is convolved with the measurements? If so, what is the form of the kernel?

4. Spatial autocorrelation results from Moran’s index: These results are not very clear to me. What is an outlier station and how does that imply the presence of spatial autocorrelation. This aspect can be discussed in the methods section. Can the authors report the numerical values of the index? It is difficult to conclude about the presence/absence of spatial autocorrelation from Figure 4. 

5. Semi-variograms and Figure 5: The authors present semi-variograms to demonstrate the presence/absence of spatial autocorrelations. I suggest that a description of semi-variograms be provided in the methods section. Additionally, it would help if the scales of y-axis are consistent for both subfigures of Figure 5. That would improve the visualization of differences in spatial autocorrelation between Grids 2 and 11. Furthermore, can the authors explain what do the labels under the subfigures (e.g., “Model : 0.043216*Stable(1963.4,2)”) mean?

6. Conclusions: The authors make three statements about different aspects that are “confirmed” by the study. The study shows that using the new method leads to slightly different results at best (requiring a precision of two decimal points). Whether the results are better or not has not been shown conclusively given that the differences are minor, and considering the concern detailed in comment 1. The conclusions need to be softened.

7. Use of the phrase “geostatistical models”:

Note that geostatistical models include kriging-type models and should be listed as part of spatial statistical models. On lines 105 and 189, the authors have combined land surface and geostatistical models as one type of model. If the authors do not mean kriging-type approaches, then this should be checked and updated with a more appropriate phrase. 

8. Author contributions: Please revise this statement. There are three authors listed in the study, and the statement should reflect contributions of all authors.

8. While labeling your various plots, it would help if you used a phrase different than “Block Kriging”. The use of “Block Kriging” makes it seem like you are using the default block kriging approach, and not the new approach that you propose in the study. An abbreviation of Thiessen polygon-based Block Kriging (such as TBP Kriging) might be helpful.

9. line 312: equation (6) is referred to as a “theorem”. That is incorrect. A theorem is a mathematical statement that is proven. Equation (6) is simply an upscaling model.

10. An overall proofreading is needed. While the manuscript is easy to read in general, there are a few minor issues that can be ironed out. There is improper use of conjunctions (e.g., words like however, whereas) and missing/repeated words at a couple places. This sometimes makes the sentences confusing to read. Here are a couple of examples:

- start of section 2.4: it is not immediately clear that you are defining block kriging.

- line 274: “Gaussian kernel” at the start of the sentence

- line 552: replace “whereas” with “where”

- make sure acronyms are expanded when they are first mentioned (e.g., “RMSD” in the abstract)

Round 2

Reviewer 1 Report

Thank you very much for your efforts to reply my questions and comments. The manuscript is largely improved.

------------------------------------------------------------------------------

4.  Line 157

       5-minute

       Please show local standard time and the difference between the local time and local standard time, and possibility to pick up nearest data.

Response: Can you please explain what you mean? The 5-minute interval means they take a measurement every five minutes.

------------------------------------------------------------------------------

What I care is the possibility to pickup in situ data closest to the satellite flyovers. The in situ data are usually recorded by using local standard

time, Central for Texas, or local daylight saving time. The satellite flyovers of SMAP, about 06:00 and 18:00 are truly local times, UTC+longitude/15. TN and TX have same local standard time but quite different local times.

Though these are not crucial information, they will help the readers to

correctly understand your works.

Author Response

  1. Line 157

       5-minute

       Please show local standard time and the difference between the local time and local standard time, and possibility to pick up nearest data.

Response: Can you please explain what you mean? The 5-minute interval means they take a measurement every five minutes.

------------------------------------------------------------------------------

What I care is the possibility to pickup in situ data closest to the satellite flyovers. The in situ data are usually recorded by using local standard time, Central for Texas, or local daylight saving time. The satellite flyovers of SMAP, about 06:00 and 18:00 are truly local times, UTC+longitude/15. TN and TX have same local standard time but quite different local times.

Though these are not crucial information, they will help the readers to correctly understand your works.

Response:

We agree that there is a difference between SMAP’s and TxSON’s time systems. We really appreciate the reviewer for this information.

We added the longitude information in line 157, and explained the time difference between the two measurements (line 174 to line 176).

Reviewer 2 Report

I appreciate the efforts made to revise the manuscript. I have two comments on the revision:

1.     For equation 6, please express the equation in mathematical form. It currently looks like code. Also, please provide a reference for this expression of gaussian weight.

2.     Spatial autocorrelation results from Moran’s index (Figure 4 and Figure A1): It seems that the authors misinterpreted my comment. Specifically, it is not clear to me what does it mean for a station to be called an “outlier”. What does it mean that a station is detected as spatial autocorrelation? My understanding of the use of Moran’s index is that the index is computed on a spatial field, and index values are obtained for the entire spatial field as a whole. It seems the authors are doing something else which could benefit with some clarification. Also, what is the difference between red and blue colored stations?

Author Response

I appreciate the efforts made to revise the manuscript. I have two comments on the revision:

  1. For equation 6, please express the equation in mathematical form. It currently looks like code. Also, please provide a reference for this expression of gaussian weight.

Response: Thank you for this comment, we updated the equation and added two references (references 46 and 47).

  1. Spatial autocorrelation results from Moran’s index (Figure 4 and Figure A1): It seems that the authors misinterpreted my comment. Specifically, it is not clear to me what does it mean for a station to be called an “outlier”. What does it mean that a station is detected as spatial autocorrelation? My understanding of the use of Moran’s index is that the index is computed on a spatial field, and index values are obtained for the entire spatial field as a whole. It seems the authors are doing something else which could benefit with some clarification. Also, what is the difference between red and blue colored stations?

Response: We really appreciate the reviewer for pinpointing this critical missing component in the interpretation of the index. There are two versions of Moran’s I, one is as the reviewer commented, it reports the spatial field as a whole, and the other one we used in our study is the Anselin Local Moran's I, which we denoted in line 230. This index reports which points are considered spatial autocorrelation (as reported in one form of the four: a cluster of high values (HH), a cluster of low values (LL), an outlier in which a high value is surrounded primarily by low values (HL), or an outlier in which a low value is surrounded primarily by high values (LH)). By proofreading, we found that we missed some key points about the interpretation of Local Moran’s I. Here is the list of changes we made regarding this issue:

We emphasize that we are using Anselin Local Moran's I (line 250), and we added an explanation about how we use outliers as evidence of spatial autocorrelation with this index, in section 2.3, lines 251-255;

We revised the presentation of the results of Moran’s I, in section 3.1, from lines 401 to 404;

We also revised the explanation of red and blue color in figure 4’s description (lines 418-419): we added “where red represents a high-low outlier, blue represents a low-high outlier”.